# Kinetics: Rethinking Test-Time Scaling Laws

**Ranajoy Sadhukhan**[*]  **Zhuoming Chen**[*]  **Haizhong Zheng**   **Yang Zhou**
**Emma Strubell**    **Beidi Chen**
Carnegie Mellon University, Pittsburgh, PA
{rsadhukh,zhuominc,haizhonz,yangzho6,estrubel,beidic}@andrew.cmu.edu

## Abstract

We rethink test-time scaling laws from a *practical efficiency* perspective, revealing that the effectiveness of smaller models is significantly overestimated. Prior work, grounded in compute-optimality, overlooks critical memory access bottlenecks introduced by inference-time strategies (e.g., Best-of-$N$, long CoTs). Our holistic analysis, spanning models from 0.6B to 32B parameters, reveals a new *Kinetics Scaling Law* that better guides resource allocation by incorporating both computation and memory access costs. *Kinetics Scaling Law* suggests that test-time compute is more effective when used on models above a threshold (14B) than smaller ones. A key reason is that in TTS, attention, rather than parameter count, emerges as the dominant cost factor. Motivated by this, we propose a new scaling paradigm centered on *sparse attention*, which lowers per-token cost and enables longer generations and more parallel samples within the same resource budget. Empirically, we show that sparse attention models consistently outperform dense counterparts, achieving over **60 point** gains in low-cost regimes and over **5 point** gains in high-cost regimes for problem-solving accuracy on *AIME* and *LiveCodeBench*. These results suggest that sparse attention is essential for realizing the full potential of test-time scaling because, unlike training, where parameter scaling saturates, test-time accuracy continues to improve through increased generation.

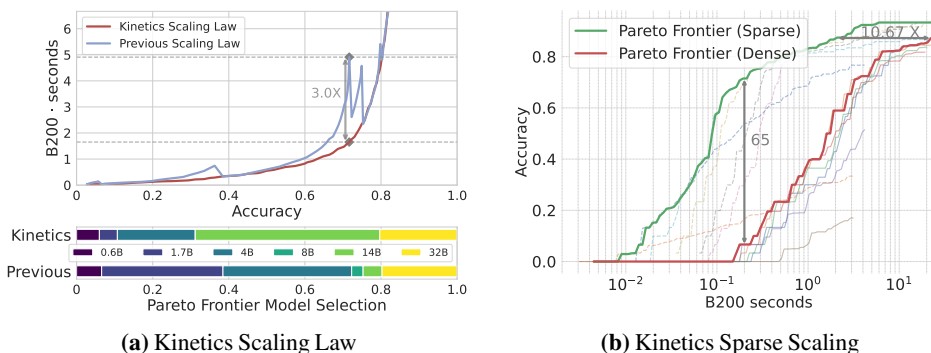

**(a)** Kinetics Scaling Law                    **(b)** Kinetics Sparse Scaling

Figure 1: **(a)**: Pareto Frontier for Qwen3 series on AIME24. Previous test-time scaling laws [4, 74, 87] focus solely on compute optimality, neglecting the significant bottleneck of memory access in long-sequence generation. This leads to suboptimal resource utilization. By incorporating memory access, the *Kinetics Scaling Law* reduces resource demands by up to $3\times$ to achieve the same accuracy. **(b)**: Inspired by the Kinetics Scaling Law, we show that *sparse attention models* scale significantly better than dense models, achieving over 50-point improvements in AIME24 in the low-cost regime and consistently outperforming dense models in the high-cost regime, in addition to substantial efficiency gains. B200 second represents the amount of work performed by a single B200 at full utilization for one second.

---

[*]Equal contribution.

39th Conference on Neural Information Processing Systems (NeurIPS 2025).

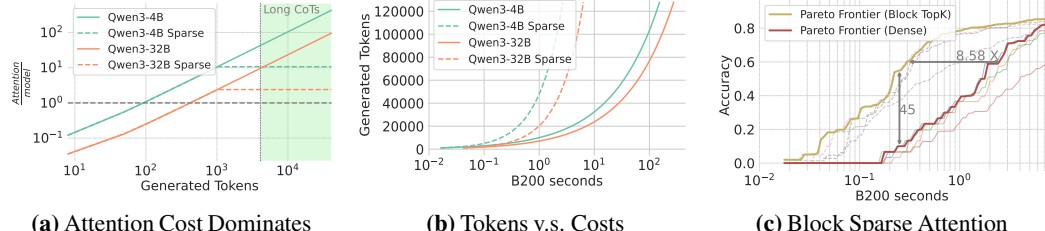

**Figure 2: (a)** Inference cost is dominated by attention, which is $100 \sim 1000\times$ more than model parameter computation, sparse attention fundamentally mitigates this bottleneck. **(b)** Under the same resource constraint, sparse attention can generate massive tokens out of the reach of dense models, which is proven to enhance the effectiveness of test-time scaling. **(c)** Simple block sparse attention yields substantial gains—improving accuracy by 45 points in the low-cost regime and achieving equivalent accuracy while using $8.58\times$ fewer resources.

## 1 Introduction

Test-time scaling (TTS) has recently emerged as a powerful strategy (e.g., Best-of-$N$, Long-CoT [86]) for enhancing the reasoning capabilities of large language models (LLMs) [30, 38, 81], particularly in scenarios where agents interact with complex environments, e.g., writing code, browsing the web [66, 92] or reinforcement learning (RL) with LLMs-in-the-loop [36, 20, 8]. These capabilities, however, introduce substantial inference-time costs, making it critical to understand performance scaling in this new paradigm. Existing scaling law studies [4, 74, 87] primarily focus on floating-point operations (FLOPs) while ignoring memory access costs, which are often the dominant factor in determining wall-clock latency in TTS regimes. As shown in Figure 1a, this gap can lead to sub-optimal deployment decisions.

In Section 3, we introduce the *Kinetics Scaling Law* for TTS, derived from a novel cost model that explicitly incorporates memory access costs. This new perspective reveals markedly different conclusions about Pareto-optimal strategies for allocating test-time compute (Figure 1a). Specifically, we find that: (1) prior scaling laws consistently **overestimate** the effectiveness of small models enhanced with inference-time strategies; and (2) computational resources are best spent first on increasing model size - up to a critical threshold (empirically around 14B parameters) - before investing in test-time strategies, such as Best-of-$N$ sampling or long CoTs. Guided by the Kinetics Scaling Law, our approach yields up to a **3×** throughput improvement on B200 hardware.

Our roofline analysis across a suite of state-of-the-art reasoning models reveals that the shift in optimal test-time compute strategies arises because test-time strategies (e.g., long CoTs, Best-of-$N$) disproportionately increase attention costs rather than parameter costs (Figure 2a). Our Iso-cost analysis shows that the quadratic growth of attention with generation length, combined with the disproportionate scaling of KV memory relative to model parameters, drives a preference for scaling up model size over generations. This imbalance is further exacerbated by MoE architectures [72, 21, 22, 1, 13, 40], which reduce active parameter count without alleviating attention overhead.

Building on this analysis, in Section 4 we introduce a new scaling paradigm, centered on **sparse** attention, which fundamentally reshapes the scaling law and significantly enhances the scalability of TTS (Figure 1b). According to our *Kinetics Sparse Scaling Law*, computational resources are best allocated to test-time strategies rather than reducing sparsity. As more computing is invested at test time, higher sparsity becomes increasingly critical to fully leveraging the benefits of these strategies. Guided by this principle, it increases problem-solving rates by up to **60** points in the low-cost regime and over **5** points in the high-cost regime on AIME24 and LiveCodeBench, through massive generated tokens, which is unaffordable for dense counterparts.

In Section 5, we demonstrate the practicality of the *Kinetics Sparse Scaling Law* using a simple block-sparse attention mechanism built on top of paged attention. This approach achieves up to a **25×** wall-clock speedup on H200 GPUs. While sparsity has traditionally been employed either for regularization in small models [82, 64] or to reduce computation in over-parameterized networks [62, 6, 32, 15, 24, 55], our work introduces a fundamentally different perspective: sparsity as a central enabler of efficient and scalable test-time inference. In contrast to pretraining – where scaling laws often exhibit diminishing returns [37] – TTS continues to benefit from increased token generation and more optimized inference paths. We hope this study can guide and encourage future co-design of model architectures, inference-time strategies, and hardware systems to fully unlock the next wave of scaling at deployment.

## 2  Related Work and Problem Settings

In this section, we first review several lines of related work relevant to *Kinetics Scaling Law*. Then we introduce a cost model accounting for computation and memory access, followed by a roofline analysis uncovering a key departure from traditional scaling laws. Finally, we outline the experimental setup used in the subsequent analysis. Notation is summarized in Table 1.

**Scaling Laws.** Prior work [42, 33, 45] has extensively examined the scaling laws of pretraining, exploring the trade-off between model size and the number of training tokens under a fixed FLOPs budget. More recently, studies such as [74, 87] have extended this analysis to test-time scaling (TTS), with a focus on compute-optimality. While these works offer a strong theoretical foundation, they largely overlook the critical bottleneck posed by memory access in current TTS systems.

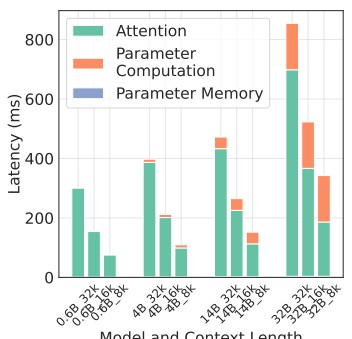

**Figure 3:** Cost Breakdown (bsz 4K).

**Test-Time Scaling.** Recent LLMs such as DeepSeek-R1 [30], OpenAI-o1/o3 [38], and QwQ [81] generate extended CoT reasoning [86] to solve complex problems, including those from AIME [59, 60]. Techniques such as parallel search through repeated sampling [4], majority voting (self-consistency) [85], and reward-model-guided inference (e.g., REBASE [87], MCTS [23, 74]) aim to improve reasoning accuracy. Strategies such as [26, 2, 80] and hybrid models [49, 69, 84] have been proposed to reduce the cost of test-time scaling.

**Sparse Attention.** A significant line of prior work has focused on overcoming the quadratic computational bottleneck of attention mechanisms during LLM training by leveraging the natural sparsity of attention matrices [11, 44, 17, 98, 3, 96]. More recently, sparse attention has experienced a resurgence in the context of LLM inference, where methods such as [101, 88, 79, 54, 10, 35] restrict the memory access of the key-value (KV) cache during generation while maintaining strong performance. These advances form a strong and steady foundation for our exploration of a new TTS paradigm.

**Table 1:** Notation Used throughout the Paper.

| Symbol | Description | Symbol | Description |
|---|---|---|---|
| $T, \mathcal{T}$ | Task (set) | $L_{out}$ | # Gen tokens |
| $M$ | Model | $N, N_T$ | Reasoning trials |
| $C, C_{\text{TTS}}(\cdot)$ | Cost function | $n, n_T$ | Max # tokens |
| $\mathcal{A}$ | Algorithm | $B, B_T$ | KV budget |
| $L_{in}$ | Prompt length | $P$ | Parameters |
| $D$ | KV size / token | $r$ | GQA ratio |

### 2.1  Cost Model

We first calculate the inference cost for the cases where the batch size is 1, and then extend to a more general case in TTS. Finally, we propose our cost model using equivalent FLOPs.

**Computation.** As discussed in [4], the computation consists of two parts: linear modules and self-attention, which is (we assume the model is served in BFloat16.)

$$C_{\text{comp}} = \underbrace{2PL_{out}}_{\text{model parameters computation}} + \underbrace{r(2L_{in} + L_{out})L_{out}D}_{\text{self-attention}}$$

**Memory Access.** Memory access also consists of two parts: model parameters and KV cache.

$$C_{\text{mem}} = \underbrace{2PL_{out}}_{\text{model parameter access}} + \underbrace{2L_{in}L_{out}D}_{\text{prompt KV cache}} + \underbrace{L_{out}^2 D}_{\text{decoding KV cache}}$$

In real serving scenarios, a large batch size will be used [18] with growing GPU VRAM [83] and model parallelism [70]. The access to the model parameter will be amortized across requests in a batch (Figure 3 shows parameter access time is negligible when the batch size is large). Thus, we only consider the second term (i.e., KV cache loading) in our cost function. Furthermore, in the cases that we have $N$ reasoning trials, the prompt cache access [41, 102] is also shared across these $N$ trials. Thus,

$$C_{\text{comp}}(N) = 2PNL_{out} + 2rNL_{in}L_{out}D + rNL_{out}^2 D \tag{1}$$

$$C_{\text{mem}}(N) = 2L_{in}L_{out}D + NL_{out}^2 D \tag{2}$$

**eFLOPs.** We propose eFLOPs (equivalent FLOPs) to capture both compute and memory access cost,
$$\text{eFLOPs} = C_{\text{comp}} + C_{\text{mem}} \times I^2 \tag{3}$$
where $I$ is the arithmetic intensity of hardware, which reflects that modern accelerators usually have a much larger computation capacity over memory bandwidth, and the gap is growing over the years [71]. In this work, we use $I = 562.5$ from NVIDIA B200 [83].

With Equations (1) to (3), we obtain the final cost model.
$$C_{\text{TTS}} = \underbrace{2NPL_{out}}_{\text{linear modules computation}} + \underbrace{2rNL_{in}DL_{out} + rNDL_{out}^2}_{\text{self-attention computation}} + \underbrace{2IL_{in}DL_{out} + INDL_{out}^2}_{\text{KV access}} \tag{4}$$
where $P, r, D$ are hyper-parameters determined by model $M$ [3].

**Roofline Analysis.** Our key insight is **attention-related cost dominates in long CoTs.** We show this by estimating the ratio of attention-related cost to parameter-related cost $\Phi$.
$$\Phi = \frac{2rL_{in}D + (rD + ID)L_{out}}{2P}$$
As shown in Figure 2a, in the regime of long CoTs, where the generation length exceeds 4096 tokens, the cost of attention surpasses that of model parameters by a factor of $100 \sim 1000$.

MLA [53] reduces KV memory access by a constant factor (similar to $r$ in GQA), it is insufficient for achieving true scalability due to several limitations: (1) MLA does not reduce attention computation; (2) the gap between FLOPs and memory bandwidth is expected to widen in the future; and (3) emerging **fine-grained MoEs** [1, 13, 40] drastically reduce FLOPs in linear layers by a factor of $10 \sim 20\times$, further increasing the relative cost of attention.

Under the context of long-CoTs being widely adopted, we can safely assume generated length $L_{out} \gg L_{in}$ or *at least* proportional to $L_{in}$. Hence, the bottleneck of inference is shifted from linear term $L_{out}P$ to the quadratic term $L_{out}^2 D$ [4].

**Experimental Setup.** In our analysis, we focus on three challenging reasoning benchmarks: AIME24 [59], AIME25 [60], math datasets spanning algebra, combinatorics, and geometry, and LiveCodeBench [39], which includes complex programming problems from recent coding competitions. We evaluate performance across various model sizes of the Qwen3 [91] series. More results are shown in the Appendix. We utilize the specs from the latest and most powerful Nvidia B200 as the basis of our theoretical studies.

## 3 Rethinking Test-time Scaling Law

In Section 3.1, we first introduce the Kinetics Scaling Law, derived from empirical investigations across the Qwen3 model series. Then, we explore the underlying reasons for the divergence between Kinetics and prior scaling laws through an Iso-Cost analysis in Section 3.2.

### 3.1 Kinetics Scaling Law

In this section, We study the scaling behavior of the Qwen3 [89, 90] considering the following problem:

*For each fixed maximum inference budget, eFLOPs per question, what is the Pareto frontier of achievable accuracy across different LLM configurations?*

With the refined cost model in Section 2.1, we first formally formulate the objective of the test-time scaling law, focusing on the tradeoff between model size and the number of generated tokens.

**Dense Scaling Law.** Given a problem instance $T$ and a total inference budget $C$, our goal is to explore the optimal tradeoff between two key factors: the choice of language model $M$, and the number of reasoning trials $N$ and the maximum generation length $n$. More precisely,
$$(N,n)_*, M_* = \arg \max_{(N,n),M} \text{Acc}(N,n,M;T) \quad \text{s.t.} \quad C_{\text{TTS}}(N,n,M;T) \leq C \tag{5}$$

---

[2]Roofline model $\max(C_{\text{comp}}, C_{\text{mem}} \times I)$ also works here and favor our claims more since most of the time $C_{\text{mem}} \times I$ dominates the cost. We choose to use an additional model because $C_{\text{comp}}$ mainly comes from linear layers while $C_{\text{mem}}$ mainly comes from the self-attention layer. The parallelization of these components during decoding remains an active area of research [103]. We discuss this roofline cost model in the Appendix.

[3]Since $L_{out}$ might differ across reasoning trials, we take the expectation for $\mathbb{E}[L_{out}]$ and $\mathbb{E}[L_{out}^2]$.

[4]This is why we call ours Kinetics Scaling Law—similar to $E_k = \frac{1}{2}mv^2$.

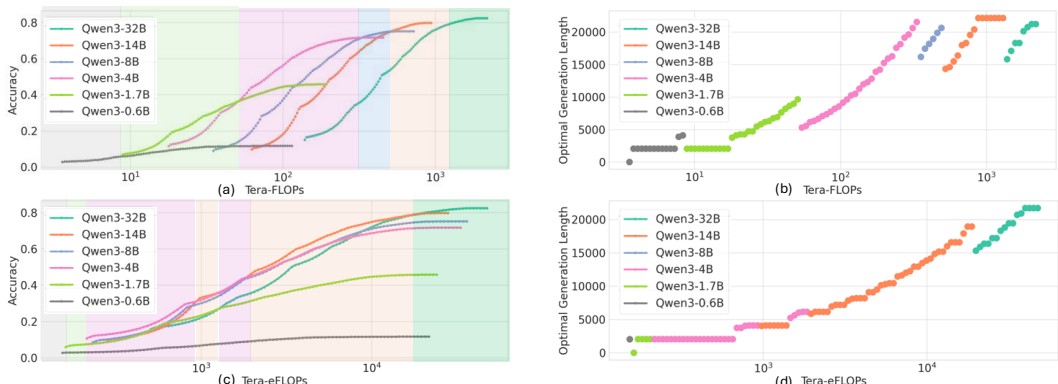

**Figure 4: AIME Pareto Frontier (Long-CoTs).** We first launch evaluations for Qwen3 series models. By controlling the maximum allowed generation lengths, we control the incurred inference cost in eFLOPs (**ab** for our scaling law) or FLOPs (**cd** for previous scaling law) and measure the accuracy (Pass@1) in AIME24. The optimal model is marked with different colors in **(ac)**. The optimal generation length is presented in **(bd)**.

Let $\mathrm{Acc}(N,n,M;T)$ denote the problem-solving rate of model $M$ on task $T$, using $N$ reasoning trials, each with a maximum reasoning length of $n$. We investigate two inference strategies: *Best-of-N* (with fixed $n$) and *Long-CoT* (with fixed $N$).

In the *Long CoTs* scenarios, where $N_T = 1$, we vary $n_T$ to evaluate the model performance under different costs. We present our results in Figure 4. Our Kinetics Scaling Law highlights two important findings compared to the previous one, which focused on merely FLOPs.

- **Efficiency of small models is overestimated.** As shown in Figures 2b and 4 **(ac)**, smaller models, despite having fewer parameters, are not as efficient as commonly assumed. For example, the 14B model outperforms both the 4B and 8B models even at low accuracy levels (e.g., below $40\%$), and the 0.6B model only lies on the Pareto frontier in regions where accuracy is negligible. In contrast, under previous scaling laws, models of all sizes span a meaningful portion of the Pareto frontier.

- **Extending CoTs are more effective than enlarging parameters only for models beyond a critical scale (empirically, 14B).** The Kinetics Scaling Law reveals that under constrained compute budgets, allocating resources to model scaling yields greater returns than increasing CoT length. As illustrated in Figure 4 **(bd)**, only the 14B and 32B models benefit from generating CoTs longer than 10K tokens; for smaller models (e.g., 1.7B and 4B), switching to a larger model is more advantageous when $L_{\mathrm{out}} < 5\mathrm{K}$. This suggests that, in practice, most of the available compute should be devoted to increasing model size rather than lengthening generations (Figure 4 **(d)**). In contrast, previous scaling laws assumed that longer CoTs provided consistent benefits across all model sizes, recommending model scaling only after CoT performance gains had plateaued.

In the *Best-of-N* setting, we fix the maximum number of generated tokens at $n_T$, and vary the number of reasoning trials $N$ to evaluate the problem-solving rate (i.e., the probability that at least one trial produces a correct answer). We have similar observations in Figures 6a to 6c. Under the previous scaling laws (Figure 6b), the most cost-effective strategy to achieve high accuracy is to apply repeated sampling using smaller models. Kinetics Scaling Law Figure 6a reveals that deploying a 14B model with fewer reasoning trials is more efficient. We also observe a critical size of 14B. For models smaller than 14B, increasing compute is best allocated toward model scaling rather than additional trials. For models at or above 14B, however, further computation is more effectively spent on increasing the number of reasoning trials, up to diminishing returns.

### 3.2 Iso-Cost Study

We attribute the above divergence between Kinetics and previous scaling laws to two reasons.

**Disproportionation between KV memory size $D$ and model parameters $P$.** Smaller models tend to require significantly more KV cache relative to their parameter size. For example, Qwen3-0.6B demands 3.5GB of KV cache to store 32K tokens, despite the model itself occupying only 1.2GB. In contrast, Qwen3-32B uses just 8GB of KV cache for the same sequence length. Empirically, doubling model parameters results in only a $1.18\times$ increase in KV cache size. As shown in Figure 5a, this phenomenon is consistently observed across model families such as OPT [100] ($1.55\times$), Qwen2.5 [90] ($1.46\times$), and LLaMA3 [27] ($1.27\times$).

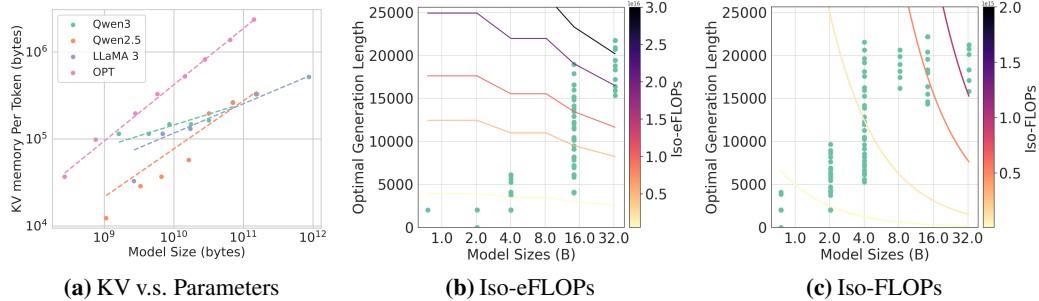

**(a)** KV v.s. Parameters        **(b)** Iso-eFLOPs        **(c)** Iso-FLOPs

**Figure 5: Explanation of the New Scaling Law. Left:** Analysis across four LLM families reveals a consistent trend of disproportionately slower KV memory growth relative to model size. For the Qwen3 series in particular, doubling model parameters results in only a $1.18\times$ increase in KV cache size. **Middle and Right:** We compare the Iso-Cost landscapes under the proposed cost model (**b**) and the traditional model (**c**).

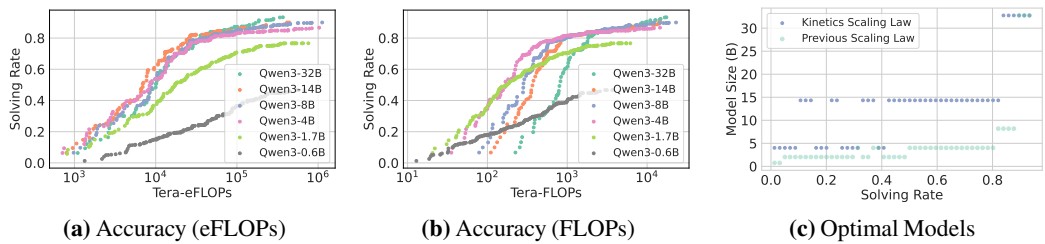

**(a)** Accuracy (eFLOPs)      **(b)** Accuracy (FLOPs)      **(c)** Optimal Models

**Figure 6: AIME Score Curve Envelope (Best-of-$N$).** We control the incurred inference cost in eFLOPs (**a**) or FLOPs (**b**) and measure the solving rate (Coverage) in AIME24 for various models by varying the maximum allowed number of reasoning trials. By taking the curve envelopes, we can project the optimal models in (**c**).

**Shift from linear to quadratic cost model.** Under this revised model, increasing generation length incurs a substantially higher cost than scaling model size; consequently, the tradeoff between model capacity and token budget shifts meaningfully. For instance, under the linear $LP$ model, the cost of generating 8K tokens with a 14B model (which is usually insufficient to solve complex tasks) is treated as equivalent to generating 24K tokens with a 4B model (sufficient to complete most tasks). However, under the $L^2D$ model, the same 14B@8K generation is only comparable in cost to a 4B@9K generation. This tighter bound makes it much harder for smaller models to compensate for their limited capacity through extended generation alone. Thus, only if the gap in model capacities is small enough (e.g., 32B only improves the accuracy by $3\%$ on AIME24 compared to 14B), the benefits of extending generation length might be more effective than directly enlarging model parameters.

Figures 5b and 5c show an Iso-Cost analysis comparing two cost models. Under Kinetics Scaling Law, the cost grows quadratically with $L_{out}$, while the KV cache scales sublinearly with model parameters $P$. As a result, when total budget is low, the Iso-eFLOPs contours tend to stretch horizontally, favoring larger model sizes over longer generation lengths. This implies that increasing model size is a more efficient use of resources than generating longer outputs. In contrast, the traditional FLOPs-based model leads to steeply vertical contours, encouraging longer generation before increasing model size.

## 4 Scale Test-time Scaling with Sparse Attention

Based on our findings in Section 3, we propose a new scaling paradigm centered on sparse attention. We begin by presenting a simple greedy algorithm for optimal resource allocation in sparse attention models, which we use to identify the Pareto frontier in Section 4.1. We then analyze the resulting changes in the scaling law and show that sparse attention models with massive TTS strategies lead to significantly higher problem-solving rates Section 4.2.

### 4.1 Optimal Resource Allocation with Sparse Attention Models

**Problem statement.** Let $\mathcal{A}$ denote the corresponding sparsity patterns (e.g., top-$k$, block sparse and local. Our goal is to explore the optimal tradeoff among three factors: model $M$, KV budget $B$, and number of trials, and the maximum generation length $(N,n)$. Specifically,

$$(N,n)_*, M_*, B_* = \arg\max_{(N,n),M,B} \text{Acc}(N,n,B,\mathcal{A},M;T)$$

$$\text{s.t.} \quad C_{\text{TTS}}(N,n,B,\mathcal{A},M;T) \leq C \quad (6)$$

The cost function $C_{\text{TTS}}$ differs from the one in Equation (4) as it incorporates sparse attention mechanisms (which makes the quadratic $L^2D$ term back to a linear term $LBD$). This modified cost model is discussed in detail in the Appendix.

**Greedy algorithm for optimal resource allocation:** We present a method to optimally schedule generation parameters $(N,n)$ and the KV budget $B$ for each task, establishing an upper bound on achievable performance and enabling analysis of the core tradeoff between TTS strategies and sparsity. We begin by solving the subproblem for each individual task $T$[5]:

$$\max \quad \text{Acc}(N_T,n_T,B_T,\mathcal{A},M;T) \quad \text{s.t.} \quad C_{\text{TTS}}(N_T,n_T,B_T,\mathcal{A},M;T) \leq C \quad (7)$$

Empirically, we discretize the searching space. For instance, in *Best-of-N*, we discretize the space of $N$ and $B$ by producing a search grid:

$$G = \{N_0,N_1,...,N_i\} \otimes \{B_0,B_1,...,B_j\}$$

For each pair $(N_a,B_b) \in G$, we compute the corresponding cost $C_{T,(a,b)}$ and accuracy $\text{Acc}_{T,(a,b)}$. We use $(N_T,B_T) \in G$ which maximizes the accuracy under the cost constraint $C$ as an approximation for Equation (7). By combining the optimal configurations $(N_T,B_T)$ for all tasks $T$, we obtain a solution to the overall problem in Equation (6). Similar discretizations also applies for *Long-CoTs*. Thus we find the optimal resource allocation.

### 4.2 Kinetics Sparse Scaling Law

Sparse attention fundamentally reshapes the Kinetics Scaling Law in Section 3 and significantly enhances the scalability of TTS. We present three important findings below.

**Sparse attention significantly enhances problem-solving performance.** As shown in Figures 8a to 8f, compared to dense baselines, for both of the inference strategies and models of various sizes, sparse attention models improve problem-solving rates by up to 60 points in the low-cost regime and over 5 points in the high-cost regime. From an efficiency perspective, dense models require over $10\times$ more eFLOPs to match the same solving rate. These findings underscore sparse attention as a key enabler for unlocking the full benefits of test-time scaling.

**Sparse attention becomes increasingly valuable in high-cost scenarios.** We investigate the tradeoff between KV budget $B$ and generation tokens $(N,n)$. For *Long-CoT*, we model the trade-offs between KV cache size, generation length, and total cost. For *Best-of-N*, we analyze how the optimal KV budget and the number of generated tokens scale with cost across $N$ reasoning trials. Our analysis reveals a consistent trend: allocating additional compute toward generating more tokens is generally more effective than expanding the KV cache. In *Best-of-N* frontier, doubling the cost leads to only a $1.18\times$ increase in KV budget, compared to a $1.74\times$ increase in total generated tokens. Similarly, for *Long-CoT*, the KV budget grows by $1.23\times$, while the number of generated tokens increases by $1.52\times$. These scaling patterns imply that as the total resource budget grows, the *sparsity ratio*, defined as the ratio of KV budget to total generated tokens, should decrease.

**Sparse attention reshapes the Kinetics Scaling Law.** As shown in Section 4.2, applying sparse attention significantly improves the efficiency of smaller models (0.6B, 1.7B, 4B), allowing them to re-emerge on the Pareto frontier across a broader range. Sparse attention reduces attention memory access from a quadratic cost term ($L^2D$) to a linear one ($LBD$), making it negligible or comparable when compared to the cost of computing with model parameters ($LP$).

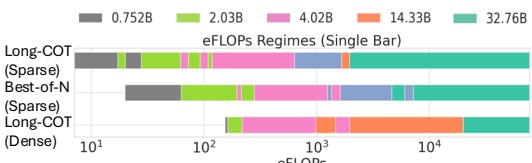

**Figure 7:** Compared to the scaling law for the dense models **(a)**, small models (0.6B, 1.7B, 4B) are more effective with sparse attention. In other words, they occupy more space in the Pareto Frontier (Figure 8a).

---

[5]For fairness, we do not schedule resources across tasks, but consider a resource upper bound for all the tasks.

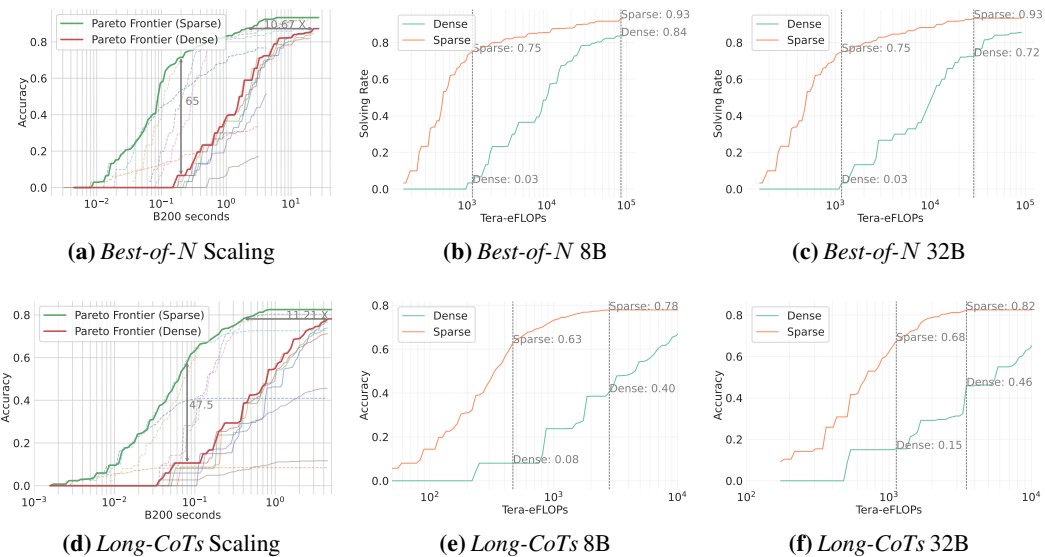

**(a)** *Best-of-N* Scaling      **(b)** *Best-of-N* 8B      **(c)** *Best-of-N* 32B

**(d)** *Long-CoTs* Scaling      **(e)** *Long-CoTs* 8B      **(f)** *Long-CoTs* 32B

**Figure 8: Sparse Attention Boosts Test-Time Scaling.** In **(a)** and **(d)**, we show that sparse attention models significantly improve the cost-accuracy trade-off under both inference strategies, ultimately achieving higher problem-solving rates at lower computational budgets. In **(b)(c)** and **(e)(f)**, we analyze individual model performance (8B and 32B) and observe that sparse attention provides notable gains. In low-cost regimes, it can enhance problem-solving rates by **50–60 percentage points**. Even in high-cost regimes, sparse models maintain an advantage of around **5 points**, while reaching these performance levels much earlier. For reference, a workload of $10^5$ Tera-eFLOPs corresponds to approximately 22 seconds of full utilization on a single B200.

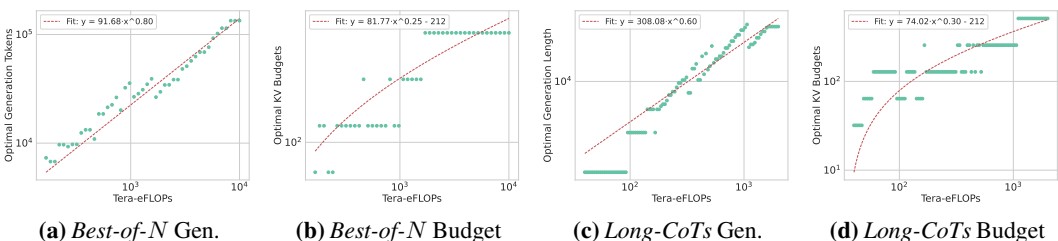

**(a)** *Best-of-N* Gen.      **(b)** *Best-of-N* Budget      **(c)** *Long-CoTs* Gen.      **(d)** *Long-CoTs* Budget

**Figure 9: Tradeoff Between Generated Tokens and KV Budget.** We empirically investigate how to balance the tradeoff between generating more tokens and allocating a larger KV cache budget, which may yield more accurate but potentially shorter outputs. Using Qwen3-8B as a representative model, we fit curves to characterize this tradeoff. For *Best-of-N*, we find that for every doubling of the total compute cost, the optimal KV budget increases by a factor of $1.18\times$, while the total number of generated tokens increases by $1.74\times$. For *Long-CoT*, the corresponding factors are $1.23\times$ and $1.52\times$, respectively. Notably, when the KV budget is small, the computational cost is dominated by model parameter-related computation rather than token generation or KV cache usage. We incorporate a model-specific constant into the cost model to account for this effect.

## 5 Experimental Validation

In this section, we demonstrate the practicality
of our sparse scaling law through block top-$k$ attention. We report empirical improvements in task throughput (number of tasks performed per unit time) using our block-sparse implementation and conduct ablation studies with alternative sparsification strategies, such as local attention, to highlight the importance of the KV selection mechanism.

### 5.1 Block Top-$k$ Attention

While top-$k$ attention offers attractive theoretical scaling, it is computationally intractable in practice. Instead, we adopt block top-$k$ attention for two key reasons. *First*, it exploits temporal locality in attention patterns [75] to retrieve semantically related key-value (KV) blocks. *Second*, its localized retrieval is hardware-efficient and integrates seamlessly with paged attention [46], enabling high-throughput decoding. In practice, we compute a representative vector for each KV block by averaging

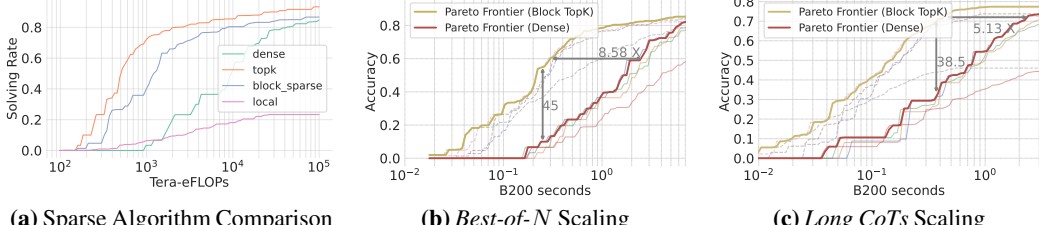

**(a)** Sparse Algorithm Comparison     **(b)** *Best-of-N* Scaling     **(c)** *Long CoTs* Scaling

**Figure 10: Sparse Attention Algorithms**. In **(a)**, we contrast the effectiveness of block top-$k$ attention against oracle top-$k$ attention and local attention. In **(b)** and **(c)**, we illustrate the optimality of block top-k sparse attention in terms of TTS on AIME24 dataset. Although upper bounded by the oracle top-k attention performance, block top-k achieves a good trade-off between effectiveness and tractability. Whereas, although easy to implement, the performance of local attention is substantially poor.

its key vectors, and use these to score the relevance of blocks to each query. Importance scores are shared across query heads within a group, following the Grouped Query Attention (GQA) scheme.

We compare block top-$k$ with local attention in Figure 10a. Although local attention is more efficient due to its static sparsity pattern, it performs significantly worse. Its poor test-time scaling prevents it from outperforming dense attention except in very low-accuracy regimes.

**Implementation**. We build our inference backend on Flashinfer [94], incorporating support for paged attention [46] and continuous batching [95]. Alongside the paged KV cache, we introduce an auxiliary data structure to store block-level average key vectors. The KV block size is chosen such that the memory load from the block-average vectors and the selected top-$k$ KV blocks remains balanced. This design enables sub-quadratic KV loading cost as the number of reasoning tokens increases.

## 5.2 Empirical Results

We quantify TTS efficiency using *task throughput*, defined as the number of tasks completed per unit time. This metric is particularly relevant for reasoning tasks, where the utility of generation hinges entirely on the correctness of the final output—unlike tasks such as summarization or content creation,

where partial outputs may still be useful. We illustrate the benefit of block top-$k$ attention across different model sizes on 8×H200 machines with an extremely large batch size of 4096. As shown in Figure 11, block top-$k$ attention substantially improves task throughput, particularly for smaller models. For instance, the Qwen3-0.6B model achieves a 12.6× to 25× increase in throughput as the generation length extends from 16k to 32k tokens. This improvement reflects the growing inefficiency of dense attention at longer contexts, which disproportionately affects smaller models. Thus, the use of sparse attention not only alleviates this bottleneck but also restores much of the practical utility of smaller models in resource-constrained settings by enabling them to use more test-time compute more cost-effectively.

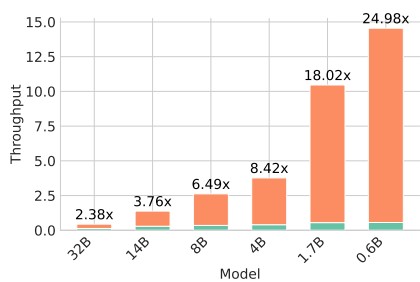

**Figure 11:** Task throughput improvement with block top-$k$ attention.

## 6 Conclusion and Discussion

This work introduces the *Kinetics Scaling Law* based on the insight that attention costs rather than parameter counts are the dominant factor in TTS. We demonstrate that sparse attention fundamentally reshapes the scaling landscape, enabling longer generations and significantly higher accuracy. We envision the Kinetics Scaling Law as a foundational tool for guiding end-to-end design across LLM serving, agent frameworks, and reinforcement learning environments. Kinetics Sparse Scaling may signal a new paradigm, enabling continued progress even beyond pretraining plateaus. While our analysis centers on NVIDIA GPUs, the underlying principle that scaling memory bandwidth is more challenging and costly than scaling FLOPs applies broadly across hardware platforms. Ultimately, this study highlights the need for co-designing model architectures, test-time inference techniques, and hardware infrastructure as a critical step toward enabling the next wave of scalable LLM deployment.

We include discussions of limitations and broader impact in the Appendix of supplementary material.

## Acknowledgements

We would like to thank Yong Wu, Xinyu Yang and Harry Dong for providing us constructive feedback on our paper and computing resources of NVIDIA. This work was partially supported by Google Research Award, Amazon Research Award, Intel, Li Auto, Moffett AI, and CMU CyLab Seed funding. This material is also based upon work supported by the National Science Foundation under Grant No. 2326610. Any opinions, findings, and conclusions or recommendations expressed are those of the authors and do not necessarily reflect the views of the National Science Foundation.

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

# Appendix

## Table of Contents

## A   Cost Model

In this section, we delve into the cost models used in the Kinetics Scaling Law. We show empirically that adopting a max cost model does not alter the scaling behavior and outline methods for calculating the cost of sparse attention models.

### A.1   Max Cost Model v.s. Additive Cost Model

Max cost model is widely used in performance modeling [97]. It assumes that computation and memory operations can be fully overlapped with each other and only considers the bottleneck operation for cost measurement.

$$C_{\text{max-cost}} = \max(C_{\text{comp}}, C_{\text{mem}} \times I)$$

where $C_{\text{comp}}$ denotes the compute cost, $C_{\text{mem}}$ the memory cost per access, and $I$ the memory intensity.

In this section, we analyze the Kinetics Scaling Law using the max cost model. For clarity, we refer to the cost model $C_{\text{comp}} + C_{\text{mem}} \times I$, which is used in the main paper, as **the additive cost model**.

We draw two conclusions from empirical results **under the max cost model**:

- **Kinetics scaling law for dense models still holds.** We re-plot Figure 4**(a)(b)** and Figure 6a under the measurement of max cost models in Figures 12 and 13. We find except that in Long-CoTs scenarios, large models become slightly more effective in low-cost regime (with accuracy$\sim$0.3), the overall trends are very close to the plots with additive cost models.

- **Sparse attention solves problems more cost-effectively.** We re-plot Figures 8a and 8d in Figures 14a and 14b. Under the max cost models, in Long-CoTs, the accuracy and efficiency gaps increase from $47.5$ points and $11.21\times$ to $52.8$ points and $15.71\times$, respectively. In Best-of-$N$, the gaps widen from $65$ points and $10.67\times$ to $69.4$ points and $19.64\times$. These results indicate that under the max cost model, our claim that sparse attention can enhance problem-solving performance is strengthen. Compared to dense attention models, sparse attention models tend to have more balanced memory and compute costs. Thus omitting one of them via a max cost model will favor sparse attention models.

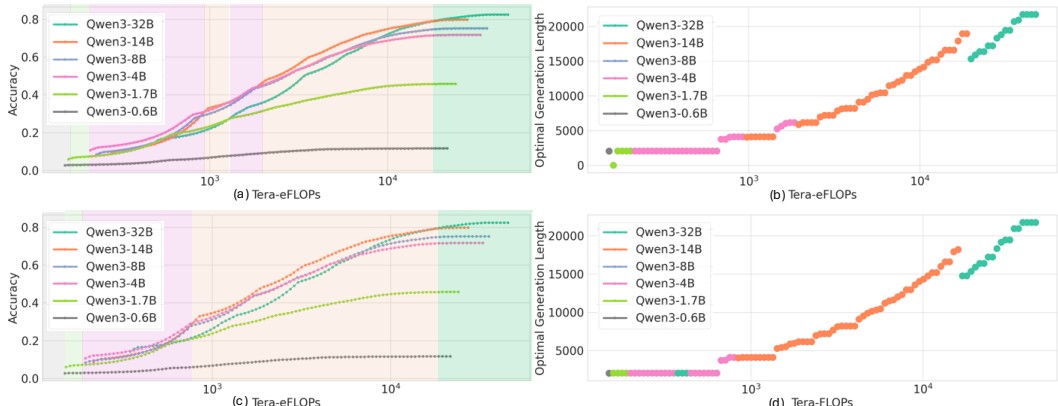

**Figure 12: AIME Pareto Frontier (Long-CoTs) with Max Cost Models. (a)(b)** is the original plot with the additive cost model. **(c)(d)** is the corresponding plot using max cost models. Compared to the original plots, the overall trend is similar except that larger models span a slightly broader region on the Pareto frontier. For example, the 14B model now consistently outperforms the 4B model with a noticeable gap around accuracy 0.3 and maintains dominance thereafter. In contrast, under the additive cost model in Figure 4**(a)**, the two models alternate in performance until accuracy exceeds 0.4. This suggests that, when evaluated using a max cost model, larger models appear slightly more efficient relative to their performance under additive cost models.

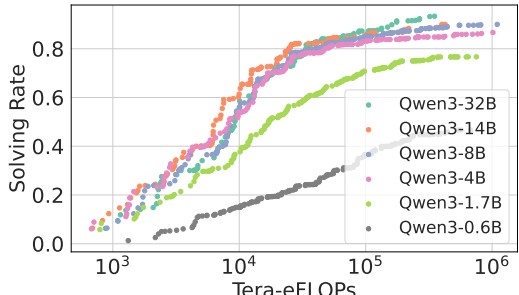

**Figure 13: AIME Pareto Frontier (Best-of-$N$) with Max Cost Models.** We re-plot Figure 6a using max cost models. The Pareto Frontier is very similar under different cost models.

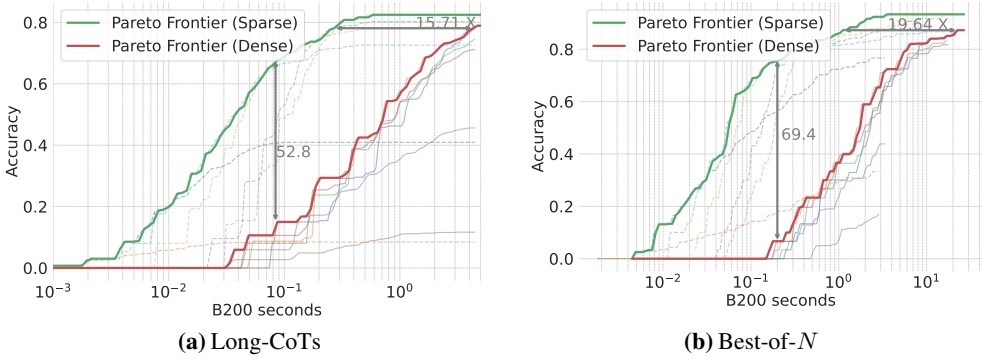

**(a)** Long-CoTs        **(b)** Best-of-$N$

**Figure 14: Sparse attention scales significantly better under max cost models.** We re-plot Figures 8a and 8d using max cost models. Compared to the original plots, the performance and efficiency gaps between sparse attention models and dense models become more pronounced. In Long-CoTs, the accuracy and efficiency gaps increase from $47.5$ points and $11.21\times$ to $52.8$ points and $15.71\times$, respectively. In Best-of-$N$, the gaps widen from $65$ points and $10.67\times$ to $69.4$ points and $19.64\times$.

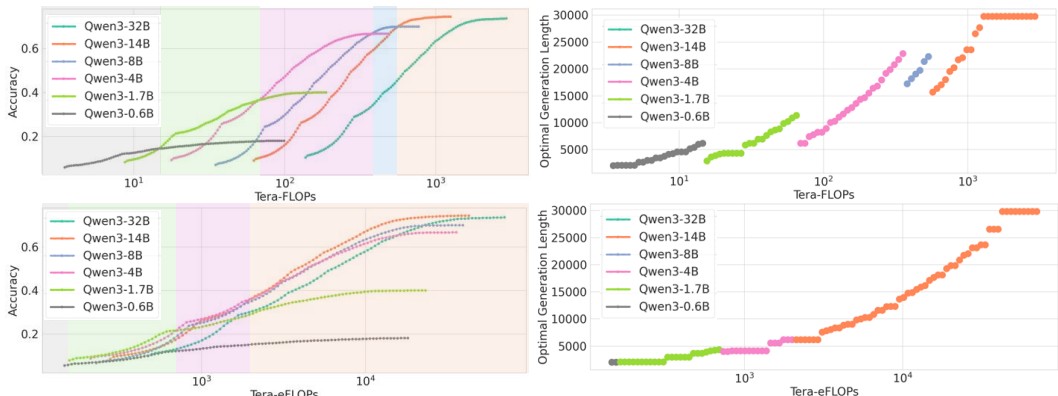

**Figure 15: AIME25 Pareto Frontier (Long-CoTs).** We conduct the same experiments as Figure 4.

## A.2  Details about Sparse Attention Cost Model

Sparse attention models follow different cost functions due to the sparsification of KV memory access. In this paper, we focus on algorithms that impose a uniform KV budget (denoted as $B$) per attention head for each decoded token. We consider $L_{in} \geq B$ for the sake of simplicity. Under this setting, the cost model for sparse attention is given by:

$$C_{\text{sparse}} = \underbrace{2NPL_{\text{out}} + 2rNDBL_{\text{out}}}_{\text{compute}} + \underbrace{2INDBL_{\text{out}}}_{\text{memory}}. \tag{8}$$

In practical implementations, we must also account for the overhead associated with retrieving or searching KV memory, denoted as $C_{\text{search}}$, which depends on the specific sparse attention algorithm $\mathcal{A}$. For example, in block top-$k$ selection, the search cost is:

$$C_{\text{search}} = \underbrace{\frac{2NL_{\text{in}}DL_{\text{out}} + rNDL_{\text{out}}^2}{2\text{Block-Size}}}_{\text{compute}} + \underbrace{\frac{2IL_{\text{in}}DL_{\text{out}} + INDL_{\text{out}}^2}{2\text{Block-Size}}}_{\text{memory}}. \tag{9}$$

In our work, we choose the Block-Size in such a way that $C_{\text{sparse}}$ and $C_{\text{search}}$ are roughly balanced, so that the sparse attention cost increases sub-linearly with generation length.

For local attention and oracle top-$k$ attention, we assume no search overhead, i.e., $C_{\text{search}} = 0$.

Many sparse attention algorithms skip the first layer [79, 10, 101], resulting in only a minor increase in total cost. For the Qwen3 series, this additional overhead is bounded by $3.57\%$ for the 0.6B model and by $1.56\%$ for the 32B model.

## B  Dense Scaling Law

In this section, we further verify Kinetics Scaling Law for dense models proposed in Section 3 with extended experimental results of different benchmarks and model series.

### B.1  Additional Benchmarks

We evaluate on AIME25 in Figures 15 and 16a to 16c and LiveCodeBench[6]in Figures 17 and 18a to 18c (excluding the 0.6B model), following the setting described in Section 3. The empirical results support the Kinetics Scaling Law: across both benchmarks, the 0.6B and 1.7B models are consistently less effective, and the Pareto frontier is almost always dominated by the 14B models.

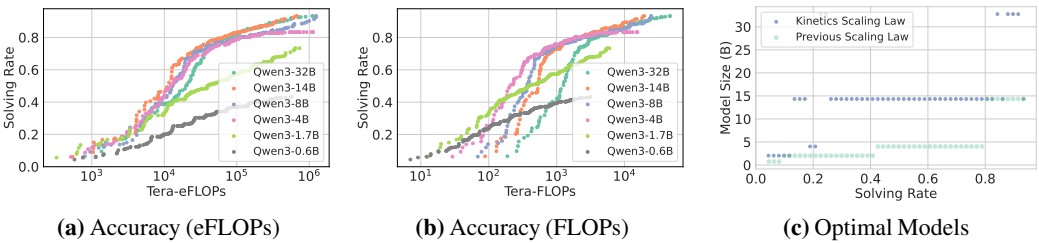

**(a)** Accuracy (eFLOPs)      **(b)** Accuracy (FLOPs)      **(c)** Optimal Models

**Figure 16: AIME25 Score Curve (Best-of-$N$).** We conduct the same experiments as Figures 6a to 6c.

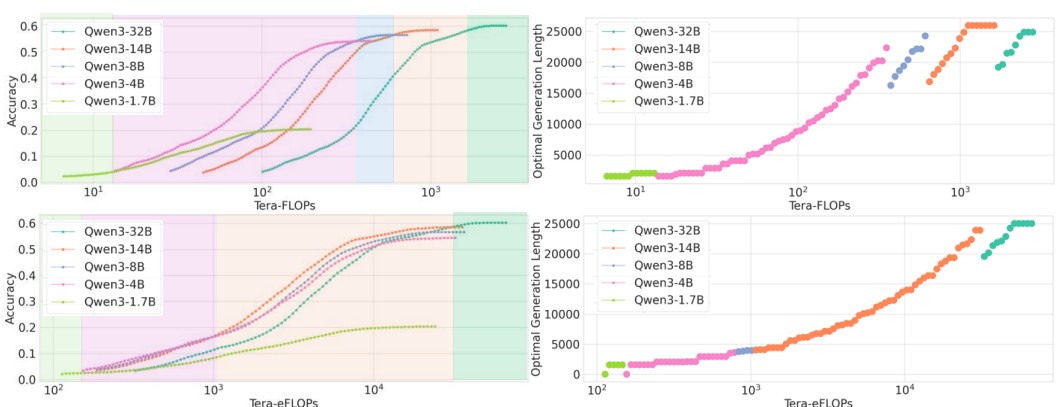

**Figure 17: LiveCodeBench Pareto Frontier (Long-CoTs).** We conduct the same experiments as Figure 4.

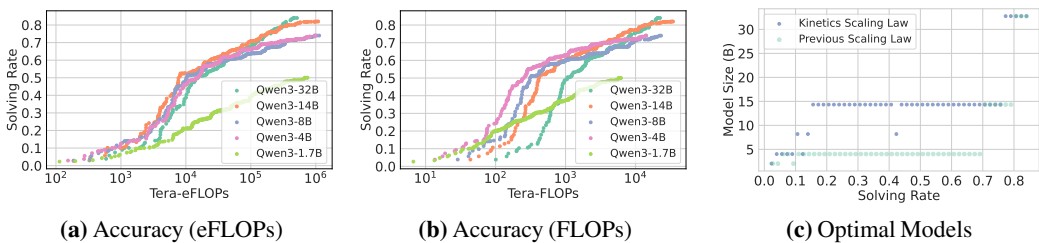

**(a)** Accuracy (eFLOPs)      **(b)** Accuracy (FLOPs)      **(c)** Optimal Models

**Figure 18: LiveCodeBench Score Curve (Best-of-$N$).** We conduct the same experiments as Figures 6a to 6c.

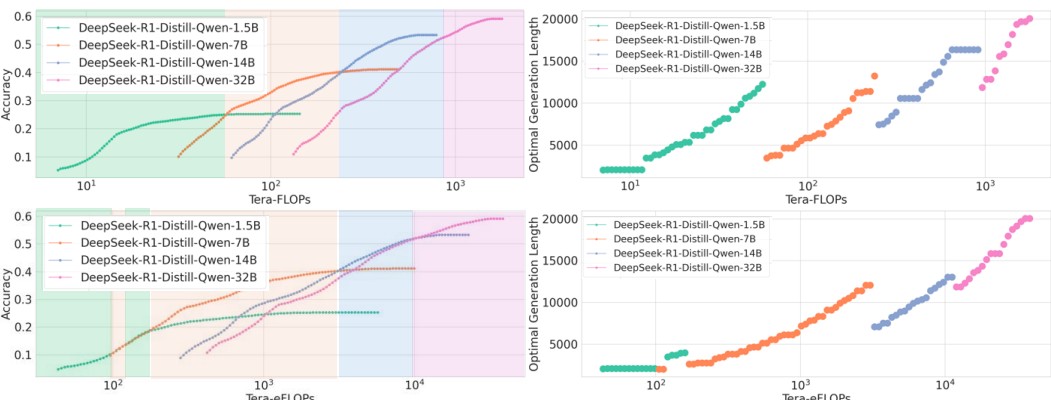

**Figure 19: AIME24 Pareto Frontier (Long-CoTs).** We conduct the same experiments as Figure 4 on DeepSeek Distilled Qwen series.

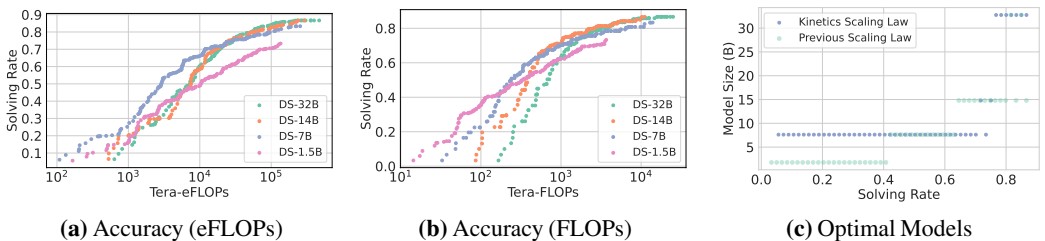

**Figure 20: AIME24 Score Curve Envelope (Best-of-$N$).** We conduct the same experiments as Figures 6a to 6c on DeepSeek Distilled Qwen series.

## B.2 Additional Reasoning Models

In Figures 19 and 20a to 20c, we evaluate DeepSeek-R1 Distilled Qwen models (abbreviated as DS models) [30] on AIME24. The DeepSeek series models further demonstrate that previous scaling laws—those based on FLOPs—significantly overestimate the effectiveness of the 1.5B model. As predicted by the Kinetics Scaling Law, increasing the number of generated tokens for the 1.5B model is less effective than scaling up the model size, such as using the 7B or larger variants.

Interestingly, we observe a shift in the emerging model size: unlike Qwen3, where the 14B model dominates, the 7B model becomes the dominant choice in the DeepSeek series. In Figures 19, 20a and 20c, the 7B model spans most of the Pareto frontier, and Figure 19 shows that 7B models with long CoTs are more efficient and effective than 14B models with short generations. We attribute this to an architectural outlier in the DeepSeek-R1 (Qwen2.5) model series. As shown in Table 2, the DeepSeek-R1 7B model is significantly more KV memory-efficient than the Qwen3-8B model. Unlike most model series illustrated in Figure 5a, where KV cache size typically grows sublinearly with respect to model parameters, DeepSeek-R1 shows a deviation from this trend: the 14B model has approximately $3.4\times$ more KV memory than the 7B model, while having only $2\times$ more parameters.

**Table 2:** KV memory Size for Qwen3 and DeepSeek-R1 Distilled models (per 32K tokens, unit: GB).

| **Qwen3** | Qwen3-1.7B | Qwen3-8B | Qwen3-14B | Qwen3-32B |
|---|---|---|---|---|
| | 3.5 | 4.5 | 6 | 8 |
| **DeepSeek** | DS-1.5B | DS-7B | DS-14B | DS-32B |
| | 0.875 | 1.75 | 6 | 8 |

This finding highlights the importance of concrete model architecture design, rather than focusing solely on the number of model parameters. Whether KV memory size is directly related to reasoning performance remains an open question, which we leave for future investigation.

## C  Sparse Scaling Law

We present additional results supporting the kinetics sparse scaling law across multiple tasks and demonstrate how these insights enable scalable test-time scaling with sparse attention.

### C.1  Additional Benchmarks

Beyond AIME24, we evaluate our approach on LiveCodeBench [39] and AIME25 [60]. Live-CodeBench features complex programming problems from recent coding contests, while AIME25 consists of challenging math problems. In both cases, sparse attention—particularly oracle top-$k$—consistently outperforms dense attention. Block top-$k$ attention, a tractable alternative, closely matches the performance of the oracle.

---

[6]For LiveCodeBench dataset, we have sampled 50 examples from the *v5* subset consisting 167 examples. Our subset comprises 24 hard, 16 medium and 10 easy examples respectively.

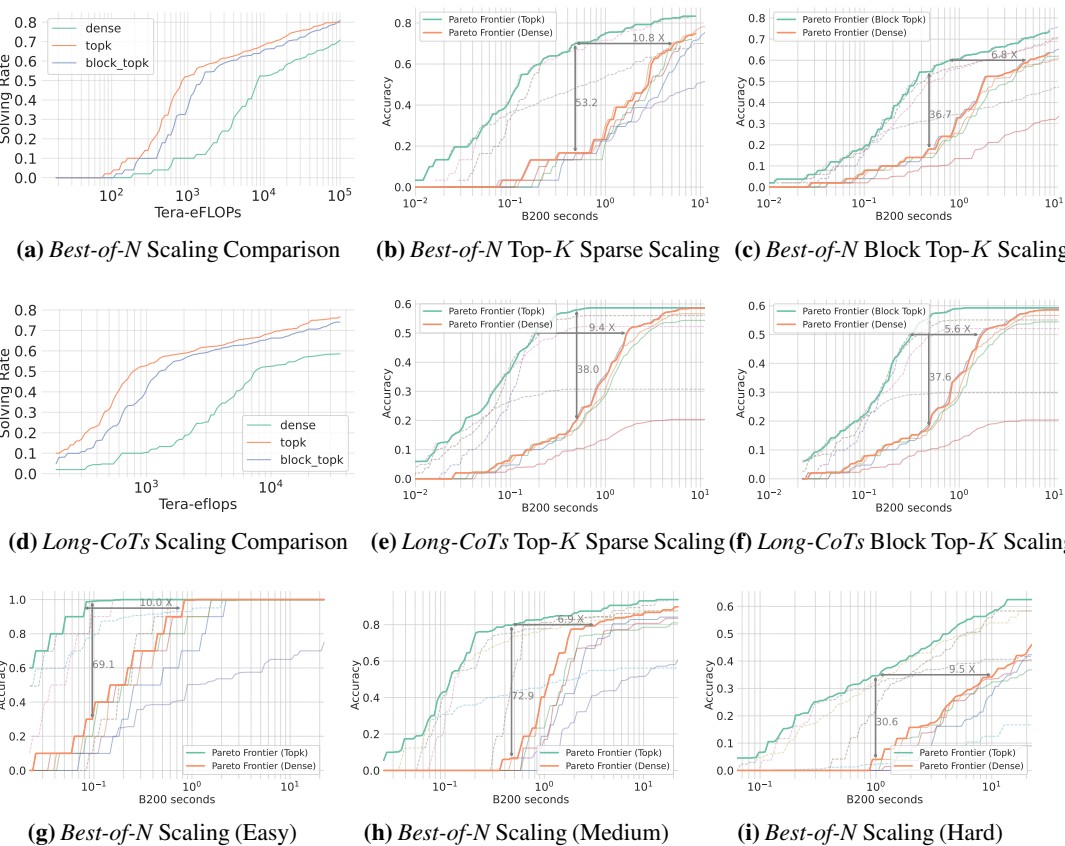

**Figure 21: LiveCodeBench Sparse Scaling.** We evaluate sparse scaling laws for Qwen3-14B model using oracle top-$k$ and block-top-$k$ attention on the LiveCodeBench dataset. **(a)(d)** compare block-top-$k$ and oracle top-$k$ with dense scaling under *Best-of-N* and *long-CoT* TTS settings. **(b)(e)** show cost-accuracy trade-offs for top-$k$ attention. **(c)(f)** show trade-offs for block-top-$k$ attention. **(g)(h)(i)** compare the oracle top-$k$ scaling for easy, medium and hard difficulty questions.

For LiveCodeBench, we sample 50 problems from the *v5* subset (24 hard, 16 medium, 10 easy). As shown in Figure 21, oracle top-$k$ attention can achieve $\sim 10\times$ speedup in high-accuracy regimes and improves coverage by $40$–$50\%$ in low-cost regimes. Conversely, the tractable alternative, Block top-$k$ yields $5$–$6\times$ speedup and $30$–$40\%$ coverage gains. We further show how the benefits of sparse attention scale with problem difficulty (Figures 21g to 21i).

Figure 22 confirms similar trends for AIME25, with substantial gains in both accuracy and efficiency under sparse attention.

## C.2 Additional Analysis

Fixing a model (e.g., Qwen3-8B), we investigate the tradeoff between generating more tokens through Best-of-$N$ and increasing the KV budget in Figures 23a to 23d. As the figures suggest, on AIME25, each doubling of total compute cost increases the optimal KV budget by $1.13\times$, while generated tokens grow by $1.67\times$; on LiveCodeBench, these factors are $1.14\times$ and $1.89\times$, respectively. We find that although the concrete numbers depend on the types of tasks, the overall results confirm our suggestions in the main paper that allocating compute toward generating more responses is generally more effective than expanding KV budget, highlighting the scalability of sparse attention.

# D  Experimental Details

In this section, we explain the details about our experiments.

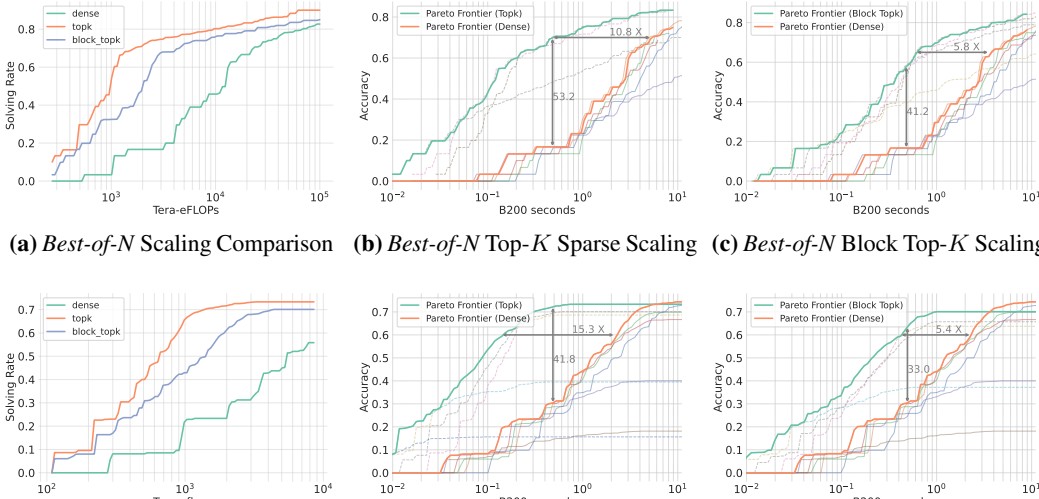

**(a)** *Best-of-N* Scaling Comparison   **(b)** *Best-of-N* Top-$K$ Sparse Scaling   **(c)** *Best-of-N* Block Top-$K$ Scaling

**(d)** *Long-CoTs* Scaling Comparison **(e)** *Long-CoTs* Top-$K$ Sparse Scaling **(f)** *Long-CoTs* Block Top-$K$ Scaling

**Figure 22: AIME25 Sparse Scaling.** We evaluate sparse scaling laws for Qwen3-14B model using oracle top-$k$ and block-top-$k$ attention on the AIME25 dataset. **(a)(d)** compare block-top-$k$ and oracle top-$k$ with dense scaling under *Best-of-N* and *long-CoT* settings. **(b)(e)** show cost-accuracy trade-offs for oracle top-$k$ attention. **(c)(f)** show trade-offs for block-top-$k$ attention.

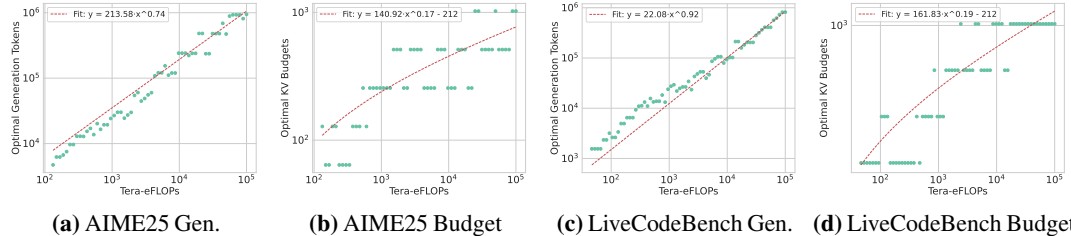

**(a)** AIME25 Gen.   **(b)** AIME25 Budget   **(c)** LiveCodeBench Gen.   **(d)** LiveCodeBench Budget

**Figure 23: Tradeoff Between Generated Tokens and KV Budget.** We empirically characterize the tradeoff between increasing generation length and allocating a larger KV cache budget using Qwen3-8B. For AIME25 (**(a)(b)**) and LiveCodeBench (**(c)(d)**), we identify the optimal KV budget and generated tokens (defined as number of reasoning trials times the average generated tokens per trial) to achieve the highest problem-solving rate under every cost constraint $C$.

## D.1   Estimate Cost, Accuracy and Solving Rate

When empirically measuring cost, one major challenge is the difficulty of controlling the actual generation length. Although it is possible to set an upper bound on the number of generated tokens, there is no guarantee that the model will utilize the full budget. For instance, in our Best-of-$N$ experiments, we set the maximum number of generated tokens to 32,768, yet the average generation length was only 14K–16K tokens.

Furthermore, it is important to model the relationship between actual inference cost and performance metrics, such as accuracy in Long-CoTs or solving rate in Best-of-$N$. Relying solely on the maximum allowed generation length to estimate cost can substantially underestimate the efficiency of models that solve problems with much shorter responses—an ability that **may** reflect higher capability.

To address this challenge, we first sample $S$ independent reasoning traces $r_1, r_2, ..., r_S$ from model $M$ on task $T$, with the maximum allowed number of tokens set to $n$. We slightly generalize Equation (4)

as:

$$C_{\text{TTS}} = 2NP\mathbb{E}[L_{\text{out}}] + 2rNL_{\text{in}}D\mathbb{E}[L_{\text{out}}] + rND\mathbb{E}[L_{\text{out}}^2]$$
$$+ 2IL_{\text{in}}D\mathbb{E}[L_{\text{out}}] + IND\mathbb{E}[L_{\text{out}}^2]$$
$$= a\mathbb{E}[L_{\text{out}}] + b\mathbb{E}[L_{\text{out}}^2] + c, \tag{10}$$

where $a$, $b$, and $c$ are constants determined by the model architecture and test-time strategies (e.g., the value of $n$). The expectations are estimated from the sampled traces, whose distribution is influenced by the model $M$, the token limit $n$, and the task $T$.

**For Long-CoTs**, we fix $N = 1$ in Equation (10) and vary $n$. From the sampled traces, we estimate the accuracy (Pass@1), and compute the corresponding cost by substituting the empirical values of $\mathbb{E}[L_{\text{out}}]$ and $\mathbb{E}[L_{\text{out}}^2]$ measured under each $n$.

**For Best-of-$N$**, we fix $n = 32{,}768$, and estimate the solving rate (Pass@$K$) following the methodology of Brown et al. [4]. The corresponding cost is then computed by substituting $N = K$ into Equation (10).

Similarly, we can estimate the cost for sparse attention models using Equations (8) and (9).

Advanced control of generation lengths is an active area of research [91, 65, 57], but it is beyond the scope of this paper.

## D.2 Greedy Algorithm for Optimal Resource Allocation

We describe the procedure for identifying optimal resource allocations and establishing the Pareto frontier for sparse attention models in Algorithms 1 and 2, as a supplement to Section 4.1. Given a fixed cost constraint $C$, we perform a grid search over key parameters: KV budgets and either reasoning trials or maximum generation lengths.

Empirically, we sweep over KV budgets {32, 64, 128, 256, 512, 1024}; reasoning trials {1, 2, 4, 8, 16, 32} (with a reduced upper limit for the 14B and 32B models to save computation time); and generation lengths {2k, 4k, 6k, 8k, 10k, 12k, 14k, 16k, 18k, 20k, 22k, 24k, 26k, 28k, 30k, 32k}.

By varying the cost constraint $C$ in Algorithms 1 and 2, we obtain the performance of sparse attention models under optimal resource allocation, as shown in Figures 8a to 8f and 10a to 10c.

It is important to note that we do not consider inter-request resource scheduling strategies, such as early stopping or dynamic reallocation across requests [26], since we aim to ensure fairness across all inputs. Instead, the cost constraint $C$ is interpreted as the maximum allowable cost per request (not the average), even if some requests achieve saturated accuracy below that threshold.

## D.3 Top-$K$ Attention and Block Top-$K$ Attention

In this section, we explain the sparse attention algorithms discussed in the main paper, namely *Top-$K$ Attention* and *Block Top-$K$ Attention*.

During the decoding phase of a large language model (LLM), the self-attention mechanism computes a weighted average of past values as follows:

$$o = \text{Softmax}\left(\frac{qK^\top}{\sqrt{d}}\right)V = wV, \quad q \in \mathbb{R}^{1 \times d}, \quad K, V \in \mathbb{R}^{n \times d}, \quad w \in \mathbb{R}^{1 \times n}, \tag{11}$$

where $d$ is the head dimension and $n$ is the context length. The key and value matrices are given by $K = [k_1, k_2, ..., k_n]$, $V = [v_1, v_2, ..., v_n]$, where each $k_i, v_i \in \mathbb{R}^{1 \times d}$ are cached from previous decoding steps.

**Top-$K$ Attention.** Top-$K$ Attention is a sparsification method where only the $K$ most relevant tokens (i.e., those with the highest attention scores) are selected to compute the output. Formally, instead of computing the full softmax, we define a sparse attention weight vector:

$$w_i = \begin{cases} \frac{\exp(s_i)}{\sum_{j \in \mathcal{I}_K} \exp(s_j)} & \text{if } i \in \mathcal{I}_K, \\ 0 & \text{otherwise,} \end{cases} \quad \text{where} \quad s_i = \frac{qk_i^\top}{\sqrt{d}}, \quad \mathcal{I}_K = \text{TopK}_K(s), \tag{12}$$

Here, $\mathcal{I}_K$ denotes the indices of the top $K$ attention scores $s_i$. By masking out the less important positions, this approach reduces the computational and memory cost of attention from $\mathcal{O}(n)$ to $\mathcal{O}(K)$, where $K \ll n$.

**Algorithm 1:** Best-of-$N$ optimal resource allocation under cost $C$

---

**Data:** Tasks $\mathcal{T}$, KV budgets $\{B_1,...,B_j\}$, trial counts $\{N_1,...,N_i\}$, cost limit $C$

**Result:** Average of maximum accuracy per task under cost $C$

1  AccumBestAcc $\leftarrow 0$   Count $\leftarrow 0$;

2  **for** *task $T$ in $\mathcal{T}$* **do**

3     **for** *KV budget $B_b$* **do**

4         Generate $S \geq \max\{N_1,..,N_i\}$ responses using $B_b$ for task $T$;

5         **for** *trial count $N_a$* **do**

6             compute cost $c_{b,a}^{(T)}$;

7             **if** $c_{b,a}^{(T)} \leq C$ **then**

8                 Compute accuracy $\text{Acc}_{b,a}^{(T)} = \text{Pass@}N_a$;

9                 **if** $Acc_{b,a}^{(T)} > BestAcc$ **then**

10                     BestAcc $\leftarrow \text{Acc}_{b,a}^{(T)}$;

11                 **end if**

12             **end if**

13         **end for**

14     **end for**

15     AccumBestAcc $+=$ BestAcc; Count $+= 1$;

16  **end for**

17  AvgBestAcc $=$ AccumBestAcc$/$Count;

18  **return** AvgBestAcc;

---

**Block Top-$K$.** Block Top-$K$ Attention is a block-level sparse attention mechanism. Instead of selecting individual tokens based on attention scores, this method selects entire blocks of tokens, thereby reducing the number of attention computations.

Specifically, assume the full sequence of $n$ keys is divided into $m = \frac{n}{\texttt{BLOCK\_SIZE}}$ consecutive blocks, each of size `BLOCK_SIZE`:

$$K = [k_1,...,k_n] \rightarrow \{K_1,K_2,...,K_m\}, \quad K_i \in \mathbb{R}^{\texttt{BLOCK\_SIZE} \times d}$$

For each block $K_i$, we first compute the average key vector:

$$\bar{k}_i = \frac{1}{\texttt{BLOCK\_SIZE}} \sum_{j=1}^{\texttt{BLOCK\_SIZE}} k_{i,j}$$

Next, we compute the attention score between the query $q$ and each block's average key:

$$s_i = \frac{q\bar{k}_i^\top}{\sqrt{d}}, \quad \text{for } i = 1,2,...,m$$

We then select the top $K' = \frac{K}{\texttt{BLOCK\_SIZE}}$ blocks based on the scores $s_i$, denoted by the index set $\mathcal{J}_{K'} = \text{TopK}_{K'}(s)$. Attention is computed only over the tokens within the selected blocks. The sparse attention weights are defined as:

$$w_i = \begin{cases} \frac{\exp(s_i)}{\sum_{j \in \mathcal{I}_K} \exp(s_j)} & \text{if } i \in \mathcal{I}_K \subseteq \text{tokens in selected blocks,} \\ 0 & \text{otherwise} \end{cases}$$

For both algorithms, $K$ is the KV budget. For GQA, we conduct an average pooling across all the query heads in a group, ensuring that the total number of retrieved key-value vectors does not exceed the allocated KV budget.

## E   Extended Related Work

**Efficient Attention.** Sparse attention [44, 99, 6, 10, 101, 88, 96, 67, 11, 48, 5] has been comprehensively studied to reduce the attention cost when processing long sequeces. In parallel, approaches like FlashAttention [16, 14] accelerate attention by maximizing hardware efficiency. To address the

**Algorithm 2:** Long-CoTs optimal resource allocation under cost $C$

**Data:** Tasks $\mathcal{T}$, KV budgets $\{B_1,...,B_j\}$, gen. lengths $\{n_1,...,n_i\}$, samples $S$, cost limit $C$
**Result:** Average of maximum accuracy per task under cost $C$

1   AccumBestAcc $\leftarrow 0$   Count $\leftarrow 0$;
2   **for** *task $T$ in $\mathcal{T}$* **do**
3     BestAcc $\leftarrow 0$;
4     **for** *gen. length $n_a$* **do**
5       **for** *KV budget $B_b$* **do**
6         Generate $S$ responses using $(B_b, n_a)$; compute cost $c_{b,a}^{(T)}$;
7         **if** $c_{b,a}^{(T)} \leq C$ **then**
8           Compute accuracy $\text{Acc}_{b,a}^{(T)} = \text{Pass@1}$;
9           **if** $Acc_{b,a}^{(T)} > BestAcc$ **then**
10            BestAcc $\leftarrow \text{Acc}_{b,a}^{(T)}$;
11           **end if**
12         **end if**
13       **end for**
14     **end for**
15     AccumBestAcc $+=$ BestAcc; Count $+= 1$;
16   **end for**
17   AvgBestAcc $=$ AccumBestAcc$/$Count;
18   **return** AvgBestAcc;

---

quadratic complexity of standard attention, researchers have also explored linear attention architectures [28, 29, 43, 12]. Additionally, quantization and low-precision methods [56, 34, 52] have been broadly applied for improving inference efficiency.

**Efficient Inference.** Orca [95], vLLM [46], and SGLang [102] are widely adopted to enhance the efficiency of LLM serving. Our analysis builds on the practical designs and implementations of these systems. In parallel, speculative decoding [47, 7, 61, 71] has been proposed to mitigate the memory-bandwidth bottleneck during LLM decoding. Additionally, model compression and offloading [19, 51, 77, 73, 25] techniques are playing a crucial role in democratizing LLM deployment.

**Efficient Test-time Strategies.** Optimizing reasoning models to generate fewer tokens has been shown to directly reduce inference-time cost [80, 2, 58]. Recent work such as CoCoNut [31] and CoCoMix [78] explores conducting reasoning in a latent space, thereby reducing decoding time. Methods like ParScale [9], Tree-of-Thoughts [93], and Skeleton-of-Thoughts [68] aim to improve efficiency by enabling parallel reasoning. Architectural innovations such as CoTFormer [63] further enhance efficiency by adaptively allocating computational resources across tokens. Efficient reward-model-based [87, 74, 76] test-time scaling algorithms are also comprehensively studied.

## F   Limitations, Future Scope, and Broader Impact

**Limitations.** Our experiments primarily focus on **Qwen3** [91] and **DeepSeek-R1-Distilled-Qwen** [30], two state-of-the-art pretrained reasoning model series, evaluated from the inference perspective. However, the effects of training and post-training strategies are not fully explored and may influence the performance gaps and robustness to sparse attention mechanisms. In addition, our cost analysis assumes a cloud-based serving environment, where computational resources are typically sufficient and large batch sizes are feasible. In contrast, local deployment scenarios, such as those using Ollama[7], often face limited VRAM where access to model parameters can dominate inference costs. Smaller models may be more appropriate in such settings, and our findings may not fully extend to these use cases.

**Future Scope.** Our sparse scaling law offers valuable insights for enriching the applications of sparse attention algorithms and the design space of test-time scaling strategies. On one hand, except for top-$k$,

---
[7] `https://github.com/ollama/ollama`

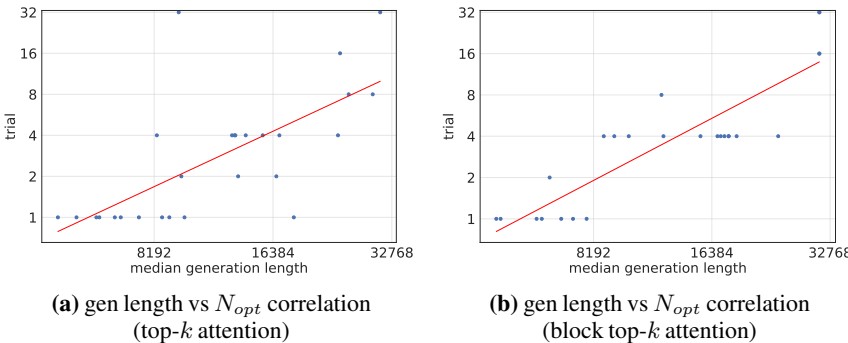

**(a)** gen length vs $N_{opt}$ correlation
(top-$k$ attention)

**(b)** gen length vs $N_{opt}$ correlation
(block top-$k$ attention)

**Figure 24: Correlation between Generation Length and Number of Trials.** Longer generations correlate strongly with the optimal number of trials ($N_{opt}$), serving as a proxy for problem difficulty. **(a)** shows this trend for top-$k$ and block top-$k$ attention on the AIME24 dataset using the Qwen3-8B model.

currently we only discuss a simple variant, i.e., block top-$k$, and have already demonstrated strong scalability. More advanced sparse attention algorithms [79, 10, 96, 50] are emerging these days. We do believe they can eventually push the scalability of test-time scaling to a much higher boundary. On the other hand, test-time scaling algorithms are proposed to adaptively allocate computation to tasks, or even to tokens [2, 63, 58, 57]. Extending them towards to new resource allocation problems in sparse attention is critical to reach the limit of Kinetics sparse scaling law. For instance, since generation length strongly correlates with the optimal number of trials under sparse attention (as shown in Figure 24), it can be used as a dynamic signal to adjust the number of trials and KV budget. Moreover, sparse attention drastically reduces inference cost, enabling more reasoning trials and longer generations. This unlocks greater flexibility in configuring TTS strategies within a fixed resource budget.

**Broader Impact.** This work aims to contribute to the understanding of efficiency and scalability challenges in the test-time scaling era, spanning model architecture, system-level implementation, and hardware design. We highlight the central role of sparsity in addressing these challenges. Our study is algorithmic in nature and does not target specific applications. While large language models can be misused in harmful ways, this work does not introduce new capabilities or risks beyond those already present in existing systems. Test-time scaling can consume a substantial amount of energy, raising concerns about the environmental sustainability of widespread deployment. By promoting sparse attention, our work hopes to help to reduce the carbon footprint and energy consumption of inference systems and support the broader goal of sustainable AI.

