# OpenReview forum: "Kinetics: Rethinking Test-Time Scaling Law"
_NeurIPS.cc/2025/Conference — NeurIPS 2025 poster_

### Official Review · Reviewer_EZZU · 2025-06-06

**Clarity:** 3
**Significance:** 4
**Originality:** 3
**Rating:** 5
**Confidence:** 4

**Summary:**

This paper challenges the prevailing test-time scaling laws that prioritize FLOPs as the sole cost metric and argues for a revised framework incorporating memory access costs. The authors introduce the Kinetics Scaling Law, showing that in inference-heavy regimes—such as Best-of-N sampling or long chain-of-thought (CoT) generation—attention costs dominate due to their quadratic complexity and KV cache overheads. Empirical studies with Qwen3 models reveal that, contrary to prior assumptions, compute is better spent on scaling model size (up to a ~14B threshold) than on increasing sample count or generation length. To improve efficiency, the authors propose a sparse attention paradigm, validated through block top-k mechanisms that significantly improve both task throughput and problem-solving accuracy, especially in compute-constrained settings.

**Questions:**

- Could the authors report results for non-Qwen models to show generality (e.g., LLaMA 3, DeepSeek, or Mixtral)?
- Does the greedy search over (N, n, B) space generalize well when real-time inference constraints (e.g., latency budgets) are added?
- How well does block top-k attention perform in settings with diverse prompt lengths or non-homogeneous input distributions?

**Ethical Concerns:**

["NO or VERY MINOR ethics concerns only"]

**Final Justification:**

The paper argues that memory access, especially KV cache traffic, dominates test-time scaling and gives a cost law for compute allocation. My initial concerns were the generality of the ~14B “critical size,” behavior under heterogeneous prompts and latency limits, coverage of sparse-attention evidence, and the search procedure. The response adds cross-GPU Pareto analyses where 14B remains on the frontier and clarifies the CPU case, which addresses generality. It also reports runs with non-shared prompts and decoding-latency tables on 8×H100 with variance, covering prompt heterogeneity and latency. Added sparsity ablations (Quest, SnapKV) support the claim that sparse attention can realize the scaling gains. The grid/greedy search with an optimality bound and notes on mapping GPU-time cost to latency reduce concern about the heuristic, though formal treatment would help. Remaining gaps are validation beyond the Qwen and DeepSeek families and missing comparisons with other sparse baselines in an end-to-end serving setup. On balance, I recommend accept.

**Quality:**

3

**Strengths And Weaknesses:**

Strengths
- The paper introduces a compelling revised cost model (Eq. 4) that includes memory access, a long-ignored but increasingly dominant factor in real-world inference.
- Empirical results are thorough, spanning multiple model sizes and datasets (AIME24, LiveCodeBench), and clearly show the dominance of attention costs (Figures 2a, 4, 6, 8).
- The proposed Kinetics Scaling Law provides actionable insight: scale model size first (up to ~14B) before investing in more sampling or longer CoTs.
- Sparse attention is not just proposed but instantiated and benchmarked through block top-k attention, showing practical gains in throughput and accuracy (Figures 10, 11).
- The iso-eFLOPs analysis is thoughtful and shows how resource allocation decisions shift once memory is included in the budget (Figure 5b vs. 5c).
- The study is grounded in realistic deployment assumptions (e.g., B200 GPUs, paged KV cache), making it highly relevant to practitioners.

Weaknesses
- The generalizability of the 14B "critical size" threshold is not fully explored across different architectures, hardware (e.g., H100, CPUs), or tasks.
- The paper evaluates sparsity patterns primarily using Qwen3 models and limited attention variants; inclusion of more baselines (e.g., LSH, SMYRF, Reformer) would strengthen the claims.
- The greedy optimization approach in Section 4.1, while reasonable, is somewhat heuristic and lacks a formal analysis or guarantees.
- Sparse attention evaluation focuses on throughput and solving rate, but ablation on latency variance or response consistency is missing.
- Some important details—like KV cache reuse in Best-of-N and Long-CoT—are deferred to the Appendix and could be brought into the main paper for clarity.

---

> ### Author Rebuttal · Authors · 2025-07-31
>
> Thank you for your thoughtful review and constructive suggestions. We are glad you found our work to offer a compelling cost model, strong empirical results, and practical insights. We have addressed your questions carefully and hope you will consider raising your score based on our response.
>
> ## W1&Q1. The generalizability of the 14B "critical size" threshold
>
> We thank the reviewer for this insightful question. In the main paper, we use Qwen3 series as an example which has 14B threshold, but the threshold varies for other model families. This threshold generalizes across GPUs (but not CPUs) and different tasks.
>
> **Architectures**
>
> As shown in Appendix B.2, **DeepSeek-Distilled-Qwen2.5 model** yields a different threshold of **7B**. However, this still supports our general conclusions,
> 1. Small models are overestimated in effectiveness.
> 2. Scaling model size up to a certain threshold can be more beneficial than increasing generated tokens. Although the threshold might vary across model families, the existence of such a threshold generalizes.
>
> **Analysis.** We further analyze the difference between the Qwen3 series and the DeepSeek-Distilled-Qwen2.5 series. The model architecture design is the key reason. The table below shows the KV memory Size for Qwen3 and DeepSeek models (per 32K tokens, unit: GB)
>
> | Qwen3 | 1.7B | 8B | 14B | 32B |
> |---|---|---|---|---|
> | KV mem | 3.5  | 4.5 | 6   | 8  |
>
> | DeepSeek | 1.5B | 7B | 14B | 32B |
> |---|---|---|---|---|
> | KV mem | 0.875 | 1.75 | 6   | 8   |
>
> As shown in the table, the DeepSeek 7B model is significantly more KV memory-efficient than the DeepSeek 14B model: the latter has  ~$3.4\times$ more KV memory, while having only $2\times$ more parameters.  While in other model series, the KV cache size typically grows sublinearly, with respect to model parameters (Figure 5a, Page 6).
>
> > As Mistral and LLaMA3 series do not have sufficient reasoning models to conduct the qualitative analysis, we do not include them. We welcome any suggestions from the reviewer for future evaluation.
>
> **Hardware**
>
> We show that 14B "critical size" generalizes to other GPUs.
>
> **Experiment Setups.** The following device-specific pareto frontiers are provided for Livecodebench. The number reported in the table is the minimum accuracy that the model appears in the Pareto Frontier. For example, the minimum accuracy in the Pareto Frontier is 17% for 14B and 59% for 32B, this suggests to achieve any accuracy in [17%, 59%), using the 14B model is most efficient. The maximum accuracy is 66%.
>
> | Long-CoT | 0.6B | 1.7B | 4B | 8B | 14B  | 32B |
> |---|---|---|---|---|---|---|
> | Prev (compute-optimal) | 0 | 0 | 4 | 55 | 57 | 59 |
> | B200 | 0 | 0 | 2 | 14 | 17 | 59 |
> | H100 | 0 | 0 | 2 | 14 | 18 | 59 |
> | A100 | 0 | 0 | 2 | 20 | 20 | 59 |
> | L40 | 0 | 0 | 2 | 14 | 19 | 59 |
>
> **Analysis.** Our scaling laws are centered on a dominant attention cost. A large gap between FLOPs and Memory I/O causes KV bottleneck to dominate, so the scaling behavior as well as the critical size remain the same.
> CPUs do not have such large FLOPs-to-I/O gap, so the 14B critical size does not apply (Appendix F, Page 19).
>
> **Task**
>
> In Appendix B.1(Page 12), we show that the 14B “critical size” can generalize to other tasks.  The number reported in the table is the minimum accuracy that the model appears in the Pareto Frontier.
>
> | Long-CoT | 0.6B | 1.7B | 4B | 8B   | 14B  | 32B  | Max Acc. |
> |---|---|---|---|---|---|---|---|
> | AIME25 (%) | 0 | 2 | 21 | 40 | 40 | 67 | 67 |
> | LiveCodeBench (%) | 0 | 0 | 2 | 14 | 17 | 59 | 66 |
>
> ## W2. Alternative sparse attention variants and models.
>
> **Sparse Attention Variants**
>
> We ablate on two sparse variants Quest[1] and SnapKV[2].
>
> > Quest requires storing the full KV cache and performs retrieval during inference. In contrast, SnapKV maintains a fixed-size KV cache. Although proposed for long-context problems, these can also be used in long-generation.
>
> Best-of-N
> | Method | B200 secs @acc=40% ↓| acc=50% ↓| acc=60% ↓| acc=70% ↓|
> |---|---|---|---|---|
> | Dense | 1.30 | 1.52 | 2.23 | 3.54 |
> | Quest | 0.21 | 0.28 | 0.38 | 0.48 |
> | SnapKV | 0.41 | 0.71 | 1.03 | 1.91 |
>
> Long-CoT
> | Method | B200 secs @acc=40% ↓| acc=50% ↓| acc=60% ↓| acc=70% ↓|
> |---|---|---|---|---|
> | Dense | 0.37 | 0.82 | 1.31 | 2.29 |
> | Quest | 0.06 | 0.14 | 0.20 | 0.31 |
> | SnapKV | 0.51 | 0.97 | 1.28 | 1.89 |
>
> **Analysis.** Quest performs better than static methods like SnapKV which requires a larger KV budget to approach the same accuracy.
>
> **Other Sparse Attention.** Sparse algorithms with better theoretical guarantees, like SMYRF[3], Reformer[4] might perform better. However, these algorithms do not have sufficient CUDA kernel support to integrate into inference systems directly. We are working on designing the kernels for these algorithms (e.g., ANNS-based algorithms) to inference efficiently. These algorithms are important to fulfill the potential of test-time computation.
>
> We will include this discussion and more experiments in the revised version.
>
> **Model Variants**
>
> **Key Results.** Similar to the Qwen3 model family, we see improved test-time scaling with sparse attention for **Deepseek-distilled models**.
>
> | Method | B200 secs @acc=40% | acc=50% | acc=60% | acc=70% |
> |---|---|---|---|---|
> | Dense | 0.45 | 0.72 | 1.46 | 3.17 |
> | Block-TopK | 0.13 | 0.23 | 0.42 | 0.57 |
>
> [1] Tang, Jiaming, et al. "Quest: Query-aware sparsity for efficient long-context LLM inference."
>
> [2] Li, Yuhong, et al. "SnapKV: LLM knows what you are looking for before generation."
>
> [3] Daras, Giannis, et al. "Smyrf-efficient attention using asymmetric clustering."
>
> [4] Kitaev, Nikita, et al. "Reformer: The efficient transformer."
>
> ## W3&Q2. Optimality of greedy search
>
> We show that the point found by our grid search consumes at most $2(1 + \epsilon)$ times more resources than the true optimal point required to reach the same accuracy, where $0 < \epsilon < 1$ and $\epsilon \to 0$ as the optimal generation length $n^*$ increases.
>
> Let the optimal configuration be $(n*, B*)$, achieving accuracy $m^*$. We perform grid search over points $(x \cdot \delta, 2^y)$, where $x, y$ are non-negative integers and $\delta$ is a positive integer. Here, $n$ is the generation length and $B$ is the KV budget.
>
> Assuming accuracy is monotonic in both $n$ and $B$, the discovered point must satisfy $(k\delta, 2^t)$ such that:
> - $(k - 1)\delta < n^* \leq k\delta$
> - $2^{t - 1} < B^* \leq 2^t$
>
> The worst-case resource overhead is:
> $$
> \frac{k\delta \cdot 2^t}{n^* B^*} < \frac{2k}{k - 1} = 2 \left(1 + \frac{1}{k - 1}\right)
> $$
>
> In practice, with $\delta = 2048$ and typically $k \gtrsim 6$, the overhead is generally bounded by $2.2 \times$ compared to the optimal point on the Pareto frontier.
>
> The greedy search over $(N, n, B)$ produces a configuration under cost $C$, which reflects the GPU time (not wall-clock time) spent on the task. To transfer to real-time latency, we can add a coefficient on $C$, to handle the case where hardware is not fully utilized or shared by other requests.
>
> ## W4. latency variance and response consistency are missing.
>
> **Latency:**
>
> **Key Results.** We report Qwen3 decoding latency using 8xH100 GPUs. The KV length is 32K tokens. The following table compares dense vs. Block Top-K attention:
> > For OOMs, we estimate full model latency using per-layer measurements. Std shown in parentheses.
>
> | Model | Dense (ms) | KV budget=1024 | 512 | 256 | 128 |
> |---|---|---|---|---|---|
> | 0.6B | 152.76 (0.03) | 14.47 (0.02) | 9.23 (0.02) | 6.24 (0.03) | 4.54 (0.02) |
> | 1.7B | 153.33 (0.03) | 15.15 (0.05) | 9.89 (0.05) | 6.84 (0.03) | 5.21 (0.03) |
> | 4B | 198.48 (0.07) | 20.59 (0.05) | 13.85 (0.04) | 10.06 (0.03) | 7.92 (0.04) |
> | 8B | 199.74 (0.04) | 21.73 (0.01) | 15.30 (0.09) | 11.35 (0.05) | 9.20 (0.04) |
> | 14B | 225.13 (0.04) | 27.77 (0.06) | 20.22 (0.01) | 15.83 (0.08) | 13.63 (0.04) |
> | 32B | 363.46 (0.17) | 47.92 (0.03) | 38.75 (0.08) | 29.15 (0.12) | 25.02 (0.08) |
>
> **Response consistency**
>
> We show that the consistency of responses is good. Maj@32 means the accuracy obtained through self-consistency[1].
>
> **Qwen3-8B**
> | Budget | Pass@32 | Maj@32 |
> |---|---|---|
> | Full | 90.0 | 83.3 |
> | 256 | 76.7 | 70.0 |
> | 512 | 83.3 | 76.7 |
> | 1024 | 90.0 | 83.3 |
>
> **Qwen3-14B**
> | Budget | Pass@32 | Maj@32 |
> |---|---|---|
> | Full | 93.3 | 83.3 |
> | 256 | 80.0 | 80.0 |
> | 512 | 86.7 | 80.0 |
> | 1024 | 93.3 | 83.3 |
>
> [1] Wang, Xuezhi, et al. "Self-consistency improves chain of thought reasoning in language models."
>
> ## W5. Important details in appendix.
>
> Thank you for the suggestion. We will bring the important details back in the main paper in the revised version.
>
> ## Q3. Diverse prompt lengths or non-homogeneous inputs?
>
> We address both efficiency and accuracy under such diverse settings.
>
> 1. Efficiency under varying prompt lengths:
> Block Top-K attention maintains strong speedups.
>
> **Key Results.** We report Qwen3 decoding latency for diverse input lengths using 8xH100 GPUs. The avg KV length is 32K with a std of 2K. The following table compares dense vs. Block Top-K attention:
>
> | Model | Dense (ms) | Block TopK budget=1024 | 512 | 256 | 128 |
> |---|---|---|---|---|---|
> | 0.6B | 153.34 (0.36) | 15.12 (0.12) | 9.58 (0.14) | 6.35 (0.19) | 4.60 (0.22) |
> | 1.7B | 154.67 (0.40) | 15.78 (0.13) | 10.20 (0.09) | 6.96 (0.11) | 5.23 (0.15) |
> | 4B | 199.91 (0.35) | 21.18 (0.15) | 14.46 (0.17) | 10.20 (0.14) | 8.00 (0.12) |
> | 8B | 200.55 (0.32) | 22.56 (0.14) | 15.69 (0.16) | 11.45 (0.17) | 9.30 (0.21) |
> | 14B | 226.88 (0.45) | 28.42 (0.18) | 20.88 (0.16) | 16.24 (0.20) | 13.47 (0.14) |
> | 32B | 365.21 (1.20) | 49.26 (0.23) | 36.37 (0.19) | 29.34 (0.24) | 24.98 (0.28) |
>
> **Explanation.** Our Flashinfer-based backend automatically partitions the workload of diverse input lengths into different pages.
>
> **2. Accuracy:** In our evaluations, different requests have different prompt lengths. Block-topk exhibits strong performance confirming its robustness across non-uniform input distributions.

---

> > ### Comment · Reviewer_EZZU · 2025-08-05
> >
> > Thank you so much for the clarifications in your rebuttal! I'm glad the feedback was helpful.
> >
> > Please let me know if any of my comments were unclear. Looking forward to seeing the final version of the paper!

---

> > > ### Author Response · Authors · 2025-08-08
> > >
> > > We sincerely appreciate your thoughtful and constructive feedback. Your suggestions are highly valuable and will contribute to improving the quality and clarity of the paper. We will make sure to incorporate them into the final version.

---

### Official Review · Reviewer_ms9j · 2025-06-29

**Clarity:** 1
**Significance:** 2
**Originality:** 3
**Rating:** 4
**Confidence:** 2

**Summary:**

This paper focus on the current test-time scaling law, and first analyze how do the performance scale in terms of eFLOPs, a new metrics that  take both computation and memory into consideration. The authors then discuss about the potential of using sparse attention to improve the current scaling law. Experiments using block top-k attention is then provided to support the advantages empirically.

**Questions:**

1. How is the results in Fig. 8 generated? Are you using the same model but with the block top-K attention introduced in Sec. 5? If so, the order of introducing the method should be changed.
2. Now that the current conclusion will be affected by the computation v.s. memory ratio, how does the results change over different GPUs?

**Ethical Concerns:**

["NO or VERY MINOR ethics concerns only"]

**Final Justification:**

My original concerns are mostly regarding clarity of the paper. The authors have promised and showed how they plan to change them. Because of the format of Neurips does not support directly change on the paper to verify the promise the author make, I cannot provide a higher rating.

**Limitations:**

yes

**Paper Formatting Concerns:**

NA.

**Quality:**

3

**Strengths And Weaknesses:**

Strength:
- The idea is relatively novel. The point of taking both compuation and memory into consideration is very interesting.
- The conclusion are supported by both theoretically and empirically.

Weakness:
- I personally encourage the authors to revise their abstract and keywords to highlight what the paper is about. I personally was thinking that certain part of training is needed for the new sparse-attention paradigm, which does not seem to be the case in here.
- While the overall paper looks well-written, certain details related to experiments are confusing. Please refer to my questions.
- The main paper seems to be over-crowded with many experimental results. I will personally recommend the authors to move some part of it to the appendix, and bring back more preliminaries, e.g., top-k attention, to make the paper more self-contained.
- While in the main paper, the authors is claiming to use three datasets, only AIME24 results are provided in the main paper, and the others are provided only in the appendix.

---

> ### Author Rebuttal · Authors · 2025-07-31
>
> Thank you very much for your thoughtful review and constructive suggestions. We are glad that the reviewer found our work to be novel and interesting, with both theoretical and empirical support. We have tried to address your questions carefully. We hope the reviewer will consider raising your score in light of our response.
>
> ## Q1. Revise the abstract and keywords to highlight what the paper is about.
>
> Thank you for raising this question. Our work studies test-time scaling law at **inference time**, focusing on the efficiency perspective. We investigate the tradeoff between models and generated tokens to reach a higher performance with a certain amount of **inference** resources. For example, to conduct a reasoning task, we can use the 32B model to generate 4K tokens, while we can also use 4B model to generate 32K tokens, but their actual cost and quality are different.
> We will emphasize that we focus on **inference** and efficiency aspects for test-time scaling in both abstracts and keywords in the revised version.
>
> **To clarify the content of our paper**, we have revised the ordering and restructured the paper. We move the definitions and preliminaries (like TopK attention) and other benchmark results (AIME25/LivecodeBench) into the main paper.
>
> For your reference, we provide a summary.
> **Summary of our paper**
> - **Objective:** We study the test time scaling law theoretically and empirically, which is about how to optimally tradeoff task performance and inference cost, *considering both computation and memory access costs*.
> - **Key contributions:**
>   - **Cost Model.** our cost model finds that attention cost dominates test-time scaling, especially for the KV memory access cost, which cannot be discovered under the previous scaling law, which only considers computation cost.  Therefore,  previously scaling law can lead to sub-optimal choices, such as overestimating small models (like 0.6B, 1.7B) with more generated tokens at test-time.
>   - **Kinetics Scaling Law.** By jointly considering memory access cost and computation cost (eFLOPs), Kinetics scaling law suggests that we need to first scale models to a certain size (14B for Qwen3) and then increase the generated tokens.
>   - **IsoCost Analysis.** In Figure 5 (Page 4), we use IsoCost curves to analyze how Kinetics scaling law and previous scaling laws are different. In Kinetics scaling law, the IsoCost curves are relatively flat along the dimension of model size and steep along the dimension of generated tokens. While in the previous scaling laws, there is an opposite trend. This suggests that, when properly considering the memory access cost, increasing CoTs is costlier than increasing model size before a certain threshold model size is reached.
>   - **Sparse Kinetics.** Based on the analysis of test-time scaling law, we propose a **new scaling paradigm with sparse attention**, which directly reduces the KV memory access, thus enabling a significant performance improvement with same inference cost (40-60 points), or reduction on inference cost  (8-10 times) to achieve the same performance.
>   - **Analysis of Sparse Kinetics.** Our analysis on Sparse Scaling Law (Fig 9., Page 8) finds that, on doubling of the total compute cost, the optimal KV budget increases by a factor of **$1.18\times$**, while the total number of generated tokens increases by **$1.74\times$** for Best of N, or **$1.23\times$** and **$1.52\times$** for Long CoTs.  This means Sparse attention becomes increasingly valuable in high-cost scenarios.
>   - **These results suggest that sparse attention is essential for realizing the full potential of test-time scaling because, unlike pretraining, where parameter scaling saturates, test-time accuracy continues to improve through increased generation.**
>
> ## Q2. How are the results in Fig. 8 generated? Are you using the same model but with the block top-K attention introduced in Sec. 5? If so, the order of introducing the method should be changed.
>
> We thank the reviewer for raising this question.  Fig. 8 (Page 8) is generated with TopK attention defined in Appendix  D.3.  In Fig. 10 (Page 9), 21 (Page 15, Appendix C.1), 22 (Page 16, Appendix C.1), we used Block-TopK attention and made a comparison between the two variants.  We have restructured the way of introducing the two variants and moved the definition of TopK attention to the main paper in our revised version.
>
>
> ## Q3. Now that the current conclusion will be affected by the computation v.s. memory ratio, how does the results change over different GPUs?
>
> We show that our conclusion holds for different GPUs for both Kinetics Scaling Law and Sparse Kinetics.
>
>
> **Kinetics Scaling Law (for dense attention models)**
>
> **Experiment Setups.** The following device-specific Pareto frontiers are provided for the Livecodebench dataset. The numbers are the ***minimum accuracy when a model appear in the Pareto frontiers***. The maximum accuracy is 66%. For example, if the minimal accuracy in the Pareto Frontier is 17% for 14B and 59% for 32B, this suggests to achieve any accuracy in [17%, 59%), using the 14B model is the most cost-effective.
>
> **Key Results.** We found that 14B models still dominate the Pareto frontiers while smaller models, like 0.6B, 1.7B only have a limited scope. We confirmed that (1)  small models' test-time scaling effectiveness is over estimated. (2) we need to scale the model first then scale the generated tokens for other GPUs as well.
>
> | Long-CoT                     | 0.6B (%) | 1.7B (%) | 4B (%)   | 8B (%)   | 14B (%)  | 32B (%) |
> |-----------------------------|------|------|------|------|------|------|
> | Previous (compute-optimal) | 0    | 0    | 4 | 55 | 57 | 59 |
> | B200                        | 0    | 0    | 2 | 14 | 17 | 59 |
> | H100                        | 0    | 0    | 2 | 14 | 18 | 59 |
> | A100                        | 0    | 0    | 2 | 20 | 20 | 59 |
> | L40                         | 0    | 0    | 2 | 14 | 19 | 59 |
>
> Our scaling laws are centered on the observation that attention cost dominates the test-time scaling cost. Therefore, if there is a large gap between FLOPs and Memory I/O in hardware (which is usually the case for modern GPUs. Despite the gaps are different for different GPUs, almost every GPU has a FLOPs/IO ratio of several hundred.), the memory access to KV cache (in attention) will become the key bottleneck,  so the scaling law curve will be very similar to Figure 1a (Page 1) in our main paper.
>
> **Sparse Kinetics (for sparse attention models)**
>
> We show that sparse attention still scales much better than dense counterparts in H100, A100, and L40.  We report the following table, which conduct the same experiments as Figure 10b (Page 9) but on configurations of different GPUs. The sparse scaling law curve remains very similar.
>
> We report the following table to show the sparse scaling law curve remains very similar to Figure 8ad (Page 8) and Figure 10 bc (Page 9).
>
> We use best-of-N scaling for the following tables.
>
> **H100**
> | Best-of-N     | H100 secs @acc=40% ↓ | acc=50% ↓ | acc=60% ↓ | acc=70% ↓ |
> |---------------|--------------------------|-----------|-----------|-----------|
> | Dense         | 3.46                     | 4.71      | 6.40      | 8.70      |
> | Block-TopK    | 0.59                     | 0.69      | 0.87      | 1.38      |
>
> **A100**
> | Best-of-N     | A100 secs @acc=40% ↓ | acc=50% ↓ | acc=60% ↓ | acc=70% ↓ |
> |---------------|--------------------------|-----------|-----------|-----------|
> | Dense         | 5.51                     | 8.08      | 10.18     | 14.94     |
> | Block-TopK    | 1.19                     | 1.38      | 1.49      | 2.55      |
>
> **L40**
> | Best-of-N     | L40 secs @acc=40% ↓ | acc=50% ↓ | acc=60% ↓ | acc=70% ↓ |
> |---------------|-------------------------|-----------|-----------|-----------|
> | Dense         | 12.97                   | 17.64     | 23.98     | 32.60     |
> | Block-TopK    | 2.40                    | 2.79      | 3.52      | 5.16      |
>
>
>  ## Q4&Q5. Over-crowded with many experimental results. Recommend to move some part of it to the appendix, and bring back more preliminaries, e.g., top-k attention, to make the paper more self-contained. Only AIME24 results are provided in the main paper, and the others are provided only in the appendix.
>
> Thank you for the suggestions. We  have (1) brought back more preliminaries in the main paper. , (2) bring other benchmarks results in the main paper,  and will (3) remove part of AIME24 results to the appendix. We hope these changes will make the paper clearer.

---

> > ### Comment · Reviewer_ms9j · 2025-08-02
> >
> > Thank you for the response. The new results on different GPUs are informative. Given that my original concerns are mostly on clarity, please do make the revision which are necessary for this paper.
> >
> > I have updated my score accordingly. Good luck!

---

### Official Review · Reviewer_cPsH · 2025-07-01

**Clarity:** 4
**Significance:** 4
**Originality:** 3
**Rating:** 5
**Confidence:** 5

**Summary:**

The authors find that small models effectiveness is overestimated in terms of inference-time scaling. Their study spans from 0.6B to 32B params, with a ‘kinetics’ scaling law, which can guide test-time compute strategies across model sizes. The key observation is that TTS requires longer context, making the FFN less of a problem and attention the major player. They propose scaling paradigms centered on sparse attn.

**Questions:**

None, well motivated paper, clean results, followed up by one potential way to alleviate the issue (Sparsity)

**Ethical Concerns:**

["NO or VERY MINOR ethics concerns only"]

**Final Justification:**

I have engaged with the authors, read the other reviewers notes and keep my vote to accept the paper.

**Limitations:**

None, in-scope experiments are sufficient, will motivate future work well.

**Paper Formatting Concerns:**

Graph formatting can be significantly improved.

**Quality:**

4

**Strengths And Weaknesses:**

Strengths
- Exposes a often overlooked observation, key is to get pareto-frontier, not the best AIME score from the smallest possible model which may take O(1k) tokens more.
- Clean results, easy to understand the core issue.

Weaknesses
- Sparse attention is good, but study on low-rank attention (KV-Cache compression) etc might be interesting. However, I treat it as orthogonal/out of scope for this paper.
- Graph formatting can be made a bit more crispy, text is way too small or light.

---

> ### Author Rebuttal · Authors · 2025-07-31
>
> Thank you very much for your thoughtful review and constructive suggestions. We are glad that you found our work to highlight an often overlooked observation, supported by clean and easily interpretable results. We have carefully addressed your questions and hope you will consider raising your score in light of our response.
>
> ## Q1: Sparse attention is good, but study on low-rank attention (KV-Cache compression) etc might be interesting. However, I treat it as orthogonal/out of scope for this paper.
>
> We thank the reviewer for this suggestion. Low rank KV compression can also effectively address the bottleneck of KV memory access and is usually **complementary** with sparse attention to further push the boundary of test-time scaling. We plan to investigate several lines of work on low rank attention into our scaling law framework.
>
> 1. Saxena, Utkarsh, et al. "Eigen attention: Attention in low-rank space for kv cache compression." arXiv preprint arXiv:2408.05646 (2024).
>
> 2. Chang, Chi-Chih, et al. "Palu: Compressing kv-cache with low-rank projection." arXiv preprint arXiv:2407.21118 (2024).
>
> 3. Zhang, Rongzhi, et al. "Lorc: Low-rank compression for llms kv cache with a progressive compression strategy." arXiv preprint arXiv:2410.03111 (2024).
>
> Also, there are some works combining sparse attention and low rank compression, for example,
>
> 4. Singhania, Prajwal, et al. "Loki: Low-rank keys for efficient sparse attention." Advances in Neural Information Processing Systems 37 (2024): 16692-16723.
>
> 5. Sun, Hanshi, et al. "Shadowkv: Kv cache in shadows for high-throughput long-context llm inference." arXiv preprint arXiv:2410.21465 (2024).
>
> 6. Yang, Shuo, et al. "Post-training sparse attention with double sparsity." arXiv preprint arXiv:2408.07092 (2024).
>
> We would be happy to hear about other suggestions.
>
> In addition, we add evaluations on SnapKV[1], a KV compression method that maintains a fixed-size KV cache by retaining only the key-value pairs identified as important by the recent tokens. This algorithm is proposed for long-context problems but can also be used in long-generation.
>
>
> **Best-of-N**
> | Method  | B200 secs for acc=40% | acc=50% | acc=60% | acc=70% |
> |---------|------------------------|---------|---------|---------|
> | Dense   | 1.30                   | 1.52    | 2.23    | 3.54    |
> | SnapKV  | 0.41                   | 0.71    | 1.03    | 1.91    |
>
> **Long CoT**
> | Method  | B200 secs for acc=40% | acc=50% | acc=60% | acc=70% |
> |---------|------------------------|---------|---------|---------|
> | Dense   | 0.37                   | 0.82    | 1.31    | 2.29    |
> | SnapKV  | 0.51                   | 0.97    | 1.28    | 1.89    |
>
> **Analysis.** SnapKV requires a larger KV budget to approach the same accuracy as Block-TopK attention as they only keep a constant memory of KV cache. However, they have an important advantage of saving GPU memory, which is essential when there is limited VRAM.
>
>  [1] Li, Yuhong, et al. "SnapKV: LLM knows what you are looking for before generation." arXiv preprint arXiv:2404.14469 (2024).
>
>
> ## Q2: Graph formatting can be made a bit more crispy, text is way too small or light.
>
> We thank the reviewer for this important suggestion. We will make the text more visible in the revised version.

---

### Official Review · Reviewer_G6EY · 2025-07-03

**Clarity:** 2
**Significance:** 3
**Originality:** 2
**Rating:** 5
**Confidence:** 3

**Summary:**

The paper challenges conventional test-time scaling laws by accounting for memory access costs, not just FLOPs, and shows how attention dominates inference costs.

The proposed “Kinetics Scaling Law” and the emphasis on sparse attention provide a fresh, impactful perspective.

Many quantitative studies show a clear analysis flow to readers.

**Questions:**

See above weakness

and

1. TTS has multiple meanings in this paper, and is hard to follow sometimes. Test-time scaling, total token sampling

2. Real-world latency benchmarks?

3. Add a brief intuitive explanation or visualization of how eFLOPs integrates compute and memory

4. The paper shows block Top-k sparse attention provides the best trade-off, but the mechanism relies on heuristics such as block averaging and grouped query attention. The KV block size and selection granularity seem critical to the performance but are not deeply analyzed.

How sensitive is the performance of your sparse models to KV block size and grouping strategy? Could this become a tuning bottleneck when transferring across tasks (e.g., from code generation to math reasoning)? Have you considered learning the sparsity patterns or integrating them with routing mechanisms like in MoEs?

**Ethical Concerns:**

["NO or VERY MINOR ethics concerns only"]

**Final Justification:**

The authors have addressed all of my concerns.

Also, I have carefully read other reviewers' comments.

Based on both of them, I decided to increase my score.

**Limitations:**

Yes

**Quality:**

3

**Strengths And Weaknesses:**

Strengths

1. Clear motivation backed by tons of quantitative studies

2. tremendous performance gains

3. Block Top-k sparse attention (Section 5) is shown to improve throughput significantly—up to 25× for small models, enabling high-efficiency inference.

Weakness

1. The scaling law threshold of 14B is specific to Qwen3 models; it’s unclear how well this generalizes to other architectures (e.g., Mistral, LLaMA3), limiting generality.

2. Want to see some real-world case studies

3. In Equation (2), the memory cost assumes prompt KV cache is amortized across reasoning trials, i.e., shared across N samples. However, in realistic settings (e.g., retrieval-augmented generation or API-style usage), prompts can vary significantly between samples.

How does the Kinetics Scaling Law behave under non-shared prompts or in settings with dynamic or partially overlapping prompts? Can your model generalize to multi-user or heterogeneous prompt settings common in LLM deployment?

---

> ### Author Rebuttal · Authors · 2025-07-31
>
> Thank you very much for your thoughtful review and constructive suggestions. We are glad that the reviewer found our work to offer a **fresh** and **impactful** perspective, with **clear motivation** and sufficient quantitative support, along with **significant performance gains**. We have tried to address your questions carefully. We hope the reviewer will consider raising your score in light of our response.
>
> ## W1: Scaling law threshold generalization
>
> We thank the reviewer for this insightful question. In the main paper, we use Qwen3 series as an example which has 14B threshold, but the threshold varies for other model families depending on their architectural details. As shown in Appendix B.2 (Page 14), our analysis of the **DeepSeek-Distilled-Qwen2.5 model** yields a different threshold of **7B**. However, this still supports our general conclusions,
> 1. Small models are often overestimated in effectiveness.
> 2. Scaling model size up to a certain threshold can be more beneficial than using more samples or longer CoTs for smaller models. Although the threshold might vary across model families depending on model architecture, the existence of such a threshold generalizes.
>
> **Analysis.** We further analyze the differences between the two model series and identify model architecture design as the key contributing factor. In particular, architectures that achieve ***slower KV memory growth across model sizes*** can shift the threshold to larger models. The table below shows the KV memory sizes for Qwen3 and DeepSeek-R1 Distilled models.
> > KV Memory size (GB) per 32K tokens
>
> | Qwen3 | 1.7B | 8B  | 14B | 32B |
> |---|----|---|---|---|
> | KV memory size | 3.5  | 4.5 | 6   | 8   |
>
> | DeepSeek Distilled | 1.5B  | 7B   | 14B | 32B |
> |---|---|---|---|---|
> | KV memory size | 0.875 | 1.75 | 6   | 8   |
>
> As shown in the table, the DeepSeek 7B model is significantly more KV memory-efficient than the DeepSeek 14B model: the 14B model has  approximately $3.4\times$ more KV memory, while having only $2\times$ more parameters.  While in other model series (LLaMa3, Qwen3, OPT), the KV cache size grows sublinearly, with respect to model parameters (Figure 5a, Page 6).
>
> > As Mistral and LLaMA3 series do not have sufficient reasoning models to conduct the qualitative analysis, we do not include them. We welcome any suggestions from the reviewer for future evaluation.
>
>
> ## W2 & Q2: Real-world latency benchmarks
>
> We implement block-topk attention with Flashinfer and torch compile.
>
> **Key Results.** We report Qwen3 decoding latency using 8xH100 GPUs. The KV Cache length is 32K tokens. The following table compares dense vs. Block Top-K attention across **different KV budgets**:
>
> > **Note**: When OOMs, we estimate total model latency using per-layer measurements.
> > *milliseconds (ms) per decoding step.*
>
> Batch Size: 1024
> | Model | Dense | KV budget=1024 | 512   | 256   | 128   |
> |---|---|---|---|---|---|
> | 0.6B  | 152.76     | 14.47        | 9.23  | 6.24  | 4.54  |
> | 1.7B | 153.33     | 15.15        | 9.89  | 6.84  | 5.21  |
> | 4B | 198.48     | 20.59        |13.85  |10.06  | 7.92  |
> | 8B  | 199.74     | 21.73        |15.30  |11.35  | 9.20  |
> | 14B | 225.13     | 27.77        |20.22  |15.83  |13.63  |
> | 32B | 363.46     | 47.92        |38.75  |29.15  |25.02  |
>
> Batch Size: 128
> | Model | Dense | KV budget=1024 | 512   | 256   | 128   |
> |---|---|---|---|---|---|
> | 0.6B | 21.2       | 4.14         | 3.39  | 3.04  | 2.16  |
> | 1.7B | 21.4       | 4.31         | 3.60  | 3.15  | 2.43  |
> | 4B | 27.8       | 5.87         | 4.94  | 4.33  | 3.82  |
> | 8B | 28.4       | 6.47         | 5.53  | 4.94  | 4.43  |
> | 14B | 32.3       | 7.85         | 6.77  | 6.24  | 5.62  |
> | 32B | 52.3       | 13.45        |11.79  |10.71  | 9.84  |
>
> Batch Size: 64
> | Model | Dense | KV budget=1024 | 512   | 256   | 128   |
> |---|---|---|---|---|---|
> | 0.6B | 11.6       | 3.46         | 3.02  | 2.97  | 2.04  |
> | 1.7B | 11.8       | 3.59         | 3.51  | 3.13  | 2.23  |
> | 4B | 15.3   | 4.91         | 4.37  | 3.95  | 3.79  |
> | 8B | 15.9 | 5.40         | 4.86  | 4.47  | 4.09  |
> | 14B | 18.2 | 6.58         | 5.90  | 5.55  | 5.08  |
> | 32B | 29.9 |11.25         |10.37  | 9.63  | 8.95  |
>
>
> ## W3: Non-shared and heterogeneous prompts generalization
>
> Yes. In summary, the Kinetics Scaling Law remains applicable in settings with non-shared/heterogeneous prompts as long as decoding remains the dominant inference cost. Moreover, our cost model naturally extends to realistic deployment scenarios, including multi-user or heterogeneous prompt settings.
>
> **Key Results.** We compare the scaling trend with both non-shared and shared prompt setting. The two resulting curves are similar.
> **Experiment Setups.** The following Pareto frontiers are provided for the Livecodebench using Best-of-N. The numbers are the minimum accuracy when a model appears in the Pareto frontiers. The maximum accuracy within our cost budget is $84\%$. e.g., the minimum accuracy in the Pareto Frontier is 16.59 for 14B in prompt-sharing setting and 77.05 for 32B, this suggests to achieve any accuracy in [16.59, 77.05), using the 14B model is most efficient.
>
> | Best-of-N | 0.6B | 1.7B | 4B   | 8B    | 14B   | 32B   |
> |---|---|---|---|---|---|---|
> | W. Sharing | 0    | 1.88 | 2.75 | 13.01 | 16.59 | 77.05  |
> | W/O. Sharing | 0    | 1.88 | 2.75 | 12.54 | 16.79 | 76.92  |
>
> **Analysis.**
>
> **Generalization to non-shared & heterogeneous prompts:** When the cost model processes non-shared prompts, we treat every single prompt as an individual request (N=1 in Equation 2). For partially overlapping prompts, we assume the cost sharing of  the “shared” part in prompts in Equation 2, for the “unshared” part, we calculate them individually for each prompt and add them up to obtain the total cost. The same logic applies to heterogeneous prompts. Thus, as long as the decoding phase dominates for each batched request, the same scaling behavior is observed.
>
> **Non-shared prompts reinforce our main insight:**  Lastly, prompt sharing is actually able to reduce the KV memory access by enabling different requests to access the same piece of KV memory at decoding time [1][2]. Without prompt sharing, the bottleneck of KV memory access will become even worse.
>
> [1] Ye, Zihao, et al. "Flashinfer: Efficient and customizable attention engine for llm inference serving."
>
> [2] vllm: [V1] [Performance] Optimize Cascade Kernel
>
> ## Q1: TTS has multiple meanings
>
> We thank the reviewer for pointing out this. Throughout the paper, we refer TTS to "Test-time-scaling" only. The “total token sampling” in Line 71 is a typo, we will fix this in the revised version.
>
>
> ## Q3: Visualization of eFLOPs
>
> We thank the reviewer for this suggestion. eFLOPs is a resource consumption metric, which reflects how much time it consumes for hardware to complete a workload at full utilization.
> If a workload consist of $A$ FLOPs for computation and $M$ bytes memory access, the time to complete the workload is modeled as $T = A/C + M/B$, where $C$ is the peak FLOPs and $B$ is the memory bandwidth. Assuming $C/B = I$,  then $T = (A + IM) / C$. Then we define ($A + IM$) as the eFLOPs.
>
> We will visualize this in the revised version.
>
> ## Q4: KV block size and grouping strategy
>
> **1. The performance is robust to KV block size and grouping strategy.**
>
> **Key Results.** We evaluate a fixed budget (1024) with different KV block sizes and grouping strategy (block-topk attention and Quest[1]). We observe that Pass@32 for different (block size, topk blocks)={(16,64), (32,32), (64,16)} are all 90% for Qwen3-8B using both block-topk and Quest.
>
> **Analysis.** With the same reasonable KV budget, the performance of Block-TopK attention and Quest (can be viewed as two grouping strategies)  is usually robust. We briefly discuss our choice of KV block size in Appendix A.2 (Page 12). We will present more ablations in the revised version.
>
> **2. The configuration of block Topk Attention is able to transfer from code generation to math reasoning.**
>
> **Key Results.** We show that a certain KV budget reaches similar relative performance on the two benchmarks.  We evaluate Pass@32 of Qwen3-8B on LiveCodeBench and AIME24.
>
> | Algo | Dense (%) | (4, 64) (budget 256) | (32, 16) (budget 512) | (16, 64) (budget 1024)|
> |---|---|---|---|---|
> | AIME24 | 90 | 76.7 | 83.3 | 90 |
> | LCB | 74 | 48 | 66 | 74 |
>
> Besides, more results on LCB are presented in Appendix C.1 (Page 14). Block TopK attention also shows impressive scaling performance as in AIME24.
>
> **3. Sparse Attention Training.**
> It is important to work on sparse attention training. Kinetics demonstrates that even inference-only algorithms (Block-TopK) can scale well. A well-trained sparse attention model is promising. However, there are some challenges:
> 1. **Optimization**: Similar to MoEs, sparse attention might be difficult to optimize. For example, sparse attention may cause signal loss in gradients, resulting in higher training loss.
> 2. **Implementation**: Efficient implementation of sparse attention kernels for training is non-trivial. FlashAttention and FlexAttention only support several constrained variants.
>
> Recent work like NSA[2] proposes a variant to solve these problems on 27B model. SeerAttention[3], MoBA[4], infLLM-v2[5] also demonstrate promising performance and will enhance the role of sparse attention in TTS. We also consider training sparse attention for test-time scaling as our important future work and are willing to discuss and receive suggestions.
>
> [1] Tang, Jiaming, et al. "Quest: Query-Aware Sparsity for Efficient Long-Context LLM Inference"
>
> [2] Yuan, Jingyang, et al. "Native sparse attention: Hardware-aligned and natively trainable sparse attention."
>
> [3] Gao, Yizhao, et al. "SeerAttention-R: Sparse Attention Adaptation for Long Reasoning."
>
> [4] Lu, Enzhe, et al. "Moba: Mixture of block attention for long-context llms."
>
> [5] Team, MiniCPM, et al. "MiniCPM4: Ultra-Efficient LLMs on End Devices."

---

> > ### Comment · Reviewer_G6EY · 2025-08-07
> >
> > Thanks for the response. Looking forward to seeing the paper and code later

---

> > > ### Author Response · Authors · 2025-08-07
> > >
> > > Thank you for your insightful and constructive comments. Your suggestions are very helpful and will certainly help improve the quality of the paper. We will definitely incorporate them into the final version and update&release the code accordingly.
> > >
> > > If you feel that your main concerns have been addressed, we would greatly appreciate it if you could kindly reconsider your initial overall evaluation.

---

### Note · Authors · 2025-08-12

We thank reviewers [R1(G6EY), R2(cPsH), R3(ms9j), R4(EZZU)] for their thoughtful and supportive feedback. We are glad they recognized the **clarity of our motivation** [R1–R4], **novelty** [R3,R4], **significance of our analysis** [R1–R4], and **strong empirical validation through extensive experiments** in realistic settings [R1,R4].
Below we summarize the major changes for the revised draft:

**[R1,R3,R4] Generalizability of Kinetics scaling law.**
We show that the *critical model size* for effective test-time scaling generalizes across GPUs and tasks. While the exact threshold varies by model family due to architectural differences, its existence is consistent.
- Explored GPUs: B200, H100, A100, L40.
- For Qwen3, the ~14B threshold holds across AIME24, AIME25, and LiveCodeBench tasks.
- For DeepSeek-distilled Qwen2.5, a similar threshold exists (7B). We explain the threshold difference in our analysis.
In the revision, we will move cross-model and cross-task studies (Appendix B.1–B.2, Pages 19–21) to the main paper and add the new hardware analysis.

**[R1,R4] Latency benchmarks and real-world case studies.**
We report decoding latency of our block-topk implementation for diverse input lengths on 8xH100 GPUs, with variance. We also show how Kinetics scaling law extends to different realistic settings such as parallel sampling with non-shared prompts and non-homogeneous requests.

**[R4] Additional sparse attention variants.**
We add ablations on Quest and SnapKV, complementing oracle top-k, block top-k, and local attention.

**[R2] Improved graph formatting.**
We shall provide cleaner, higher-resolution plots with clearer labels.

**[R3,R4] Revised abstract and paper writing.**
We will update the abstract and keywords to emphasize that our work studies *test-time scaling laws for inference* from an efficiency perspective. By jointly modeling memory and computation costs, Kinetics identifies a threshold model size before further test-time investment,contrary to prior laws,and highlights the key role of sparse attention. Key preliminaries (e.g., top-k attention, KV cache reuse in Best-of-N sampling) will be moved from the appendix to the main paper for clarity.

We believe these revisions address the reviewers’ concerns and incorporate additional suggestions to improve the work’s clarity and completeness. We thank the reviewers again for their valuable feedback, which has strengthened our paper.

---

### Decision · Program_Chairs · 2025-09-17

**Decision:**

Accept (poster)

**Comment:**

The paper received 1 borderline accept and 3 accept final ratings. Reviewers acknowledge the novelty and motivation of the paper: taking memory I/O into the scaling law consideration and improve attentions with sparse attention. Reviewers raised concerns about the writing and generalization of the method, etc. Those questions were largely addressed during the rebuttal session, one reviewer raised the score to accept (from borderline accept). AC recommends to accept the paper.